# Neutrophil extracellular traps-inspired DNA hydrogel for wound hemostatic adjuvant

Rui Ye[1,7], Ziyu Zhu[2,3,7], Tianyi Gu[2,7], Dengjie Cao[1], Kai Jiang[1], Qiang Dai[4,5], Kuoran Xing[3], Yifan Jiang[3,6], Siyi Zhou[2], Ping Cai[1], David Tai Leong ●[3] ✉, Mengfei Yu[2] ✉ & Jie Song ●[1,4] ✉

Severe traumatic bleeding may lead to extremely high mortality rates, and early intervention to stop bleeding plays as a critical role in saving lives. However, rapid hemostasis in deep non-compressible trauma using a highly water-absorbent hydrogel, combined with strong tissue adhesion and bionic procoagulant mechanism, remains a challenge. In this study, a DNA hydrogel (DNAgel) network composed of natural nucleic acids with rapid water absorption, high swelling and instant tissue adhesion is reported, like a band-aid to physically stop bleeding. The excellent swelling behavior and robust mechanical performance, meanwhile, enable the DNAgel band-aid to fill the defect cavity and exert pressure on the bleeding vessels, thereby achieving compression hemostasis for deep tissue bleeding sites. The neutrophil extra-cellular traps (NETs)-inspired DNAgel network also acts as an artificial DNA scaffold for erythrocytes to adhere and aggregate, and activates platelets, promoting coagulation cascade in a bionic way. The DNAgel achieves lower blood loss than commercial gelatin sponge (GS) in male rat trauma models. In vivo evaluation in a full-thickness skin incision model also demonstrates the ability of DNAgel for promoting wound healing. Overall, the DNAgel band-aid with great hemostatic capacity is a promising candidate for rapid hemostasis and wound healing.

Massive bleeding from vital organ wounds that can occur during major surgical procedures, traumatic accidents, gun, weapons and explosion wounds if not arrested can quickly escalate into serious outcomes including hemorrhagic shock, organ failure and death[1–5]. Conventional hemostatic approaches, such as surgical suture, tourniquet and compression often cannot stop the bleeding of deep-seated non-compressible traumas in a timely manner due to restricted accessibility and large wound area covering a large vascular network[6–8]. Recently, pre-gelation liquids while being able to enter into the wound space quickly, unfortunately suffers from poor gelation speed from the equally fast bleeding and dilution below the critical gelation concentration within the bleeding wound[9,10]. Sponges and cloth-based dressings that are even extensively used back in World War I, have very limited absorption ability and do not arrest the bleeding sufficiently to allow the

[1]Institute of Nano Biomedicine and Engineering, Department of Instrument Science and Engineering, School of Electronic Information and Electrical Engineering, Shanghai Jiao Tong University, Shanghai 200240, China. [2]The Affiliated Hospital of Stomatology, School of Stomatology, Zhejiang University School of Medicine, and Key Laboratory of Oral Biomedical Research of Zhejiang Province, Hangzhou 310006 Zhejiang, China. [3]Department of Chemical and Biomolecular Engineering, National University of Singapore, 4 Engineering Drive 4, Singapore 117585, Singapore. [4]Hangzhou Institute of Medicine, Chinese Academy of Sciences, Hangzhou 310022 Zhejiang, China. [5]College of Materials Science and Engineering, Zhejiang University of Technology, Hangzhou 310014 Zhejiang, China. [6]Department of Ultrasound in Medicine, The Second Affiliated Hospital of Zhejiang University School of Medicine, Zhejiang University, Hangzhou 310009, China. [7]These authors contributed equally: Rui Ye, Ziyu Zhu, Tianyi Gu. ✉e-mail: cheltwd@nus.edu.sg; yumengfei@zju.edu.cn; songjie@him.cas.cn

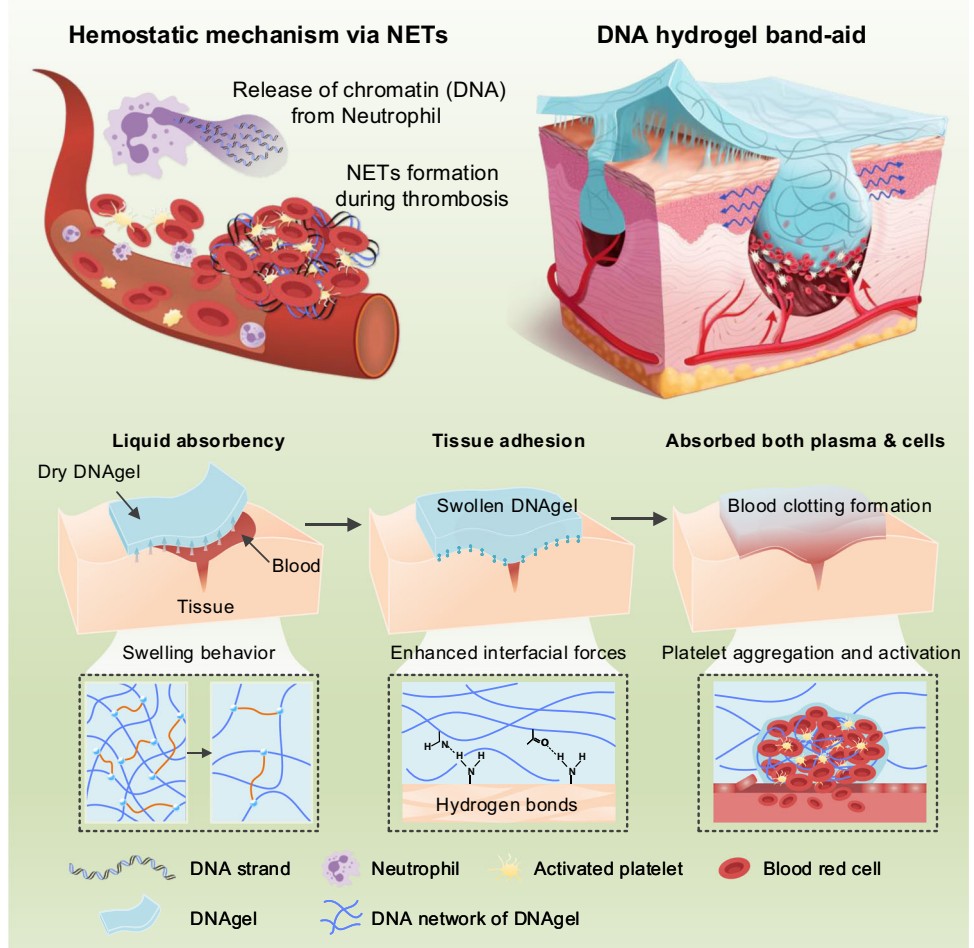

**Fig. 1 | Scheme of DNA hydrogel band-aid for accelerated hemostasis that can fulfill the clinical requirements of stabilizing a profuse bleeding wound.** Physiologically, neutrophils at the wound site released extracellular DNA to form traps (NETs) that promote fast and stable thrombosis. Similarly, our form factor conforming DNA hydrogel was able to form NETs-like thrombosis. Besides the conformational advantage, DNAgel is also highly absorptive of blood at the wound edge and high adhesive to the wetted wound surfaces, thus forming both a physical blockade and biological clot to stop the bleeding.

body's own clotting system to take over[11,12]. While the incorporation of procoagulant agent mimics the natural physiological response to bleeding, the wound scenario is clearly too overwhelming that requires external intervention to create that hemostasis event and that would still require a materialistic substrate to stem the large macroscopic bleeding[13–16] As we are also concerned about post hemostasis events, it is also vital to ensure the biocompatibility of these materials and understand their interaction with the host immune system for their successful clinical integration[17–19]. Though considerable progress has been made in hemostatic agents, developing a new biomaterial strategy that combines high swelling, strong adhesion and natural hemostatic mechanism is highly desirable for hemostasis while still taking into account the subsequent healing processes.

DNA-based materials, particularly DNA hydrogels have attracted wide attention in the recent decades, due to the advantages of intrinsic biological properties including unique sequence programmability, structural predictability as well as biocompatibility[20–23]. DNA hydrogels show tremendous potential for biomedical applications in drug delivery, cell capture and delivery, 3D cell culture, and tissue regeneration[24–27]. One important hint that further reinforced the idea of using DNA hydrogels was inspired from neutrophil extracellular traps (NETs) which is a meshwork of extracellular DNA released and precipitated fibers combined with histones and other proteins which robustly promoted coagulation at the bleeding sites[28–30]. NETs are so stable that the final network of red blood cells (RBCs), platelets, and

DNA is strong enough to form thrombi, even when fibrin is removed from the meshwork[31]. From these naturally occurring NETs, we learnt that DNA networks possess the basic building blocks for perfected biocompatibility, correct porosity to effectively quickly slow down the fast-flowing blood cells in a bleeding event to eventually immobilize the clotting components to close proximity and interactions to trigger a positively reinforcing feedback loop of clotting cascade, bringing into close proximity the various clotting components to induce a stable clot.

In this work, we utilize the naturally extracted DNA from salmon sperm, to construct a DNA hydrogel (DNAgel) network with preferable properties, including liquid absorbency, tissue adhesion and absorption both plasma and cells, as shown in Fig. 1. The hemostatic mechanism of DNAgel is simultaneously analyzed in this study. We find that the liquid absorption capacity and mechanical properties may make the DNAgel as a barrier against blood flow. Moreover, the in vitro/ in vivo hemostatic properties of DNAgel are evaluated on rat trauma models, especially on non-compressible hepatic hemostasis. The adhesion and aggregation ability of erythrocytes and activation capacity of platelets are also assessed by blood cells adhesion and flow cytometry. The macroscopic adhesiveness of the band-aid also brings together the cut edges and promote post hemostasis healing. Overall, this study provides a promising strategy for hemostatic biomaterials that arrest a massive bleeding event and de-escalates the otherwise lethality potential of the bleeding wound.

## Results

### Preparation of DNAgel

To validate this concept, we utilized biomass DNA, specifically the commercial salmon sperm DNA, to prepare DNAgel by chemical crosslinking, according to previous work[32]. New covalent bonds were formed between the amine group from the nucleosides of the DNA and the α,β-unsaturated aldehydes acrolein of the crosslinker PEGDA based on the aza-Michael addition mechanism[33], thus resulting in a DNAgel with three-dimensional (3D) network (Fig. 2a). It can be clearly observed that the DNA solution precursor was crosslinked after the introduction of PEGDA, and was able to absorb water, swell and hold the water after the addition of deionized water (DW). The representative photographs of crosslinking and swelling processes were shown in Fig. 2b. The scanning electron microscope (SEM) image (Fig. 2c) showed that the microstructures of the prepared DNAgel with porous network. The Rheology test was conducted on DNAgel to verify the gelation. As shown in the Fig. 2d, the storage modulus (G′) was higher than the loss modulus (G″) at the low strain range (0.1–10%), indicating that the DNAgel maintained a solid-like behavior. However, when the strain was larger than 193%, G′ became lower than G″, suggesting gel collapsing. Additionally, it could be interred that our DNAgel was possibly to keep the solid-like behavior at in vivo environment because the highest strain in the body have been reported not to exceed 10%[34]. The attenuated total reflection Fourier transform infrared (ATR-FTIR) spectroscopy of the DNAgel was measured. As shown in Fig. 2e, there were no significant changes in the characteristic peaks of DNA between the pure commercial salmon sperm DNA and the crosslinked DNAgel. In particular, the peaks at 3444 cm$^{-1}$ assigned to OH groups and peaks at 1225 cm$^{-1}$ assigned to the symmetric stretching vibration of phosphate groups indicated the similar hydrophilicity of DNAgel to that of DNA materials. The wettability of the DNAgel was also performed by a contact angle meter (Fig. 2f and Supplementary Movie 1) and a water droplet instantly spread over and was absorbed by the DNAgel (within 90 s).

### Swelling behavior of DNAgel

A powerful ability to absorb liquid and then expand is crucial for ideal hemostatic materials, as the pre-dried and wrinkled gel could pass through narrow wounds and expand to fill deep and irregular cavities at the bleeding site, such as gunshot wounds. Thus, the swelling behavior of DNAgel was first evaluated and shown in Fig. 2g. The swelling process of DNAgel soaked in DW was continuously monitored and it was able to absorb liquid up to 470 times its own weight (Supplementary Fig. 1). The swelling behavior reached a plateau after 8 h. Furthermore, the cylindrical DNAgel could also swell in 3D space. In particular, the DNAgel swelled in a long mid-narrowed tube, and eventually fit the shape of the tube. To determine the processability of the DNAgel, we developed several methods to obtain DNAgels in different shapes: cutting the cured DNAgel into complex patterns via masks, or shaping the precursor into different macro- and microstructures due to its in-situ gelation (Fig. 2h–j). These DNAgels retained the ability of strong water absorption and isotropic expansion.

### Adhesion of DNAgel

For deep and severe bleeding sites, the strong wet tissue adhesion endowed the gels with excellent hemostasis performance upon sealing the bleeding wound and slowing the bleeding rate. The DNAgel fabricated into flat patches (Fig. 3a) were utilized for subsequent experiments on tissue adhesion and wound hemostasis. Given the excellent water uptake capacity, the DNAgel rapidly hydrated and swelled upon contact with the wet tissue interface, driven by the negatively charged phosphate groups in the DNA strand. Simultaneously, the phosphate groups in the DNA strand formed intermolecular bonds (such as hydrogen bonds and electrostatic interactions) with the tissue surfaces (Fig. 3b). Subsequently, we quantitatively analyzed the wet shear

adhesion strength of the DNAgel according to previous research[35] and the result was exhibited in Fig. 3c and Supplementary Fig. 2. The adhesion strength of DNAgel was 5.29 ± 0.80 kPa, slightly higher than that of clinical tissue adhesive (for example, GelMA), however, significantly higher than that of control group. In addition, the results of the 90-degree peeling test (Fig. 3d) showed that the DNAgel can establish a tough adhesion with an interfacial toughness of 168.56 ± 6.82 J m$^{-2}$ between wet rat skin. To evaluate the ability of the DNAgel to adhere to wet surfaces in vivo, we also applied a rounded DNAgel to the surface of a rat liver with gentle pressure for 5 s (Fig. 3e). The DNAgel adhered firmly to the liver and was difficult to remove. Remarkably, the DNAgel disappeared when the liver was re-exposed after 7 days, as shown in Supplementary Fig. 3, suggesting that the DNAgel could be biodegraded within one week in vivo.

### In vitro hemostasis and biocompatibility of DNAgel

The capacity of aggregated erythrocytes and platelets was an essential indicator of hemostatic performance. The interaction of erythrocytes/platelets with DNAgel was observed by SEM. As shown in Fig. 4a, intact but shriveled erythrocytes adhered and piled up on the surface of the DNAgel. When the blood came to contact with the dried DNAgel, the fluid was absorbed by the DNAgel while the red blood cells (RBCs) and platelets were trapped and concentrated on the dense surfaces of the DNAgel, rather than the deeper layers (Supplementary Fig. 4). Some activated platelets with pseudopods were observed on the surface of the DNAgel. As for the gelatin sponge (GS) group, erythrocytes and platelets passed through the three-dimensional network of GS, and fewer erythrocytes were observed to attach to the smooth surface of GS. Besides, a large amount of fibrin (white arrow) was observed in the blood clot on the DNAgel surface (Supplementary Fig. 5). It was also observed by confocal laser scanning microscope (CLSM) that the RBCs (red) and platelets (green) were evenly distributed and colocalized with the three-dimensional DNA network (blue) (Fig. 4b) after DNAgel was soaked in fresh rat blood. Furthermore, platelets in activated status were found in the DNA network (blue), appearing red fluorescent (Supplementary Fig. 6). However, other cells related to blood clotting, such as lymphocytes, monocytes, and leucocytes, were barely observed. It was indicated that the DNA network may provide a scaffold for the enrichment and adhesion of RBCs and platelets, rather than other cells. The schematic illustration of adhesion and activation of erythrocytes and platelets on DNA gel was shown in (Fig. 4f). Additionally, the in vitro blood coagulation test revealed that the DNAgel could absorb blood and facilitate blood coagulation within 10 min (Fig. 4c and Supplementary Fig. 7). The blood clotting index (BCI) was also used for a preliminary assessment of the hemostatic capacity of DNAgel. As shown in Fig. 4d, the quantitative values in the DNAgel and GS groups reached 5.76 ± 1.57% and 28.82 ± 0.75%, respectively, suggesting the good blood clotting capacity of DNAgel. Besides, the biocompatibility of the DNAgel was also investigated. The in vitro viability of WRL68 cells incubated with DNAgel-conditioned medium was equivalent to that of the control medium, indicating the excellent cytocompatibility of DNAgel after 48-h culture. (Supplementary Fig. 8). On the other hand, the hemolysis assay was performed to evaluate the hemocompatibility of DNAgel. The quantitative hemolysis ratios of DNAgel at different concentrations showed no obvious variation with the PBS negative control group and commercial GS group (Fig. 4e). Additionally, rheological properties of DNAgel after fully swelling in the PBS and also the heparinized fresh whole blood were measured to ensure the stability of DNAgel in vivo (Supplementary Figs. 9, 10).

### In vivo hemostatic performance of DNAgel

Since the DNAgel had absorbent property, adhesive capacity, and the ability to adhere and accumulate erythrocytes and platelets, we assumed that the DNAgel should be a promising candidate for rapid hemostasis,

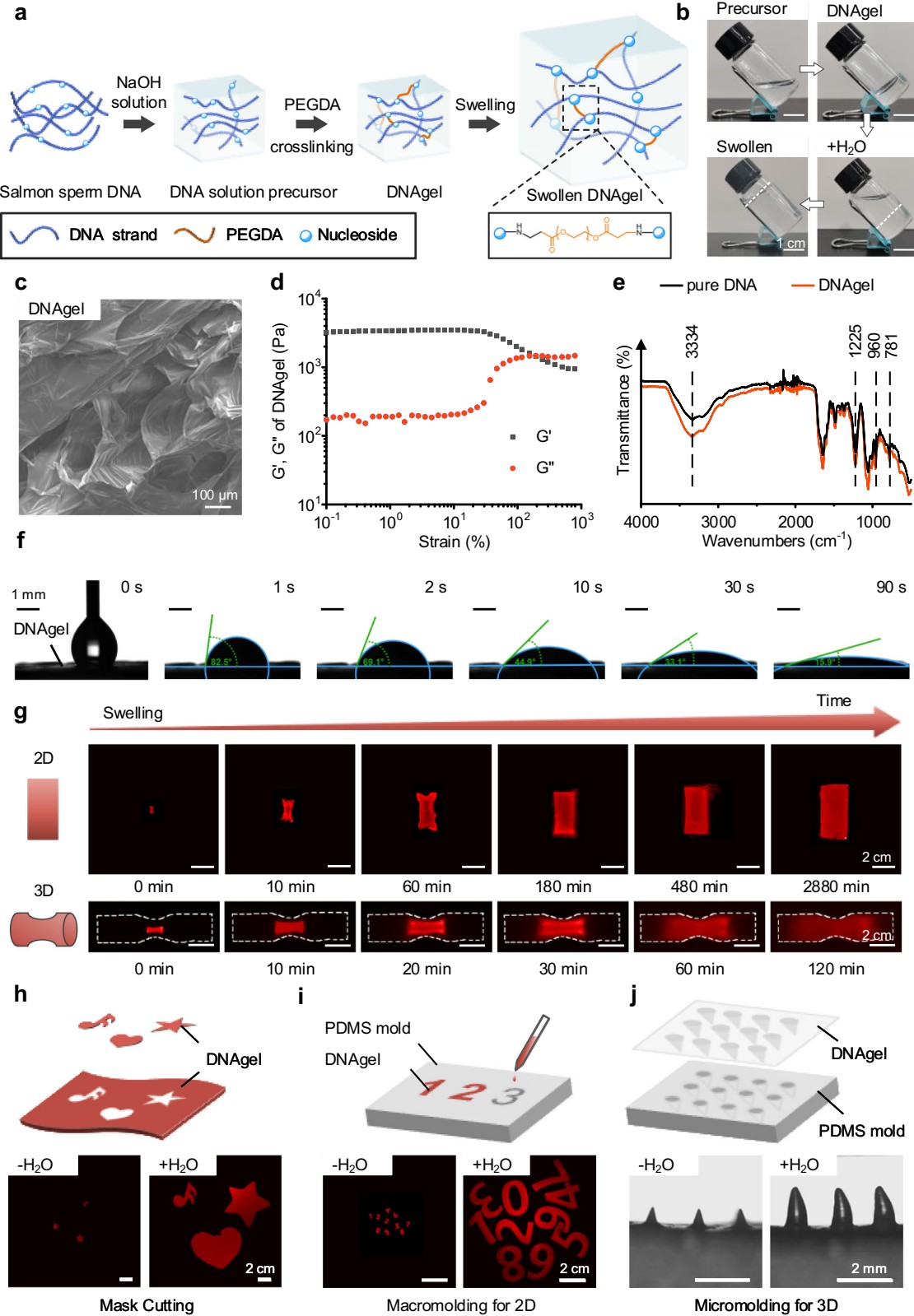

**Fig. 2 | Preparation and expansion properties of DNAgel. a** Schematic of the synthesis process. **b** Photographs of the DNA solution before and after crosslinking. Scale bar: 1 cm. **c** SEM image of DNAgel. Scale bar: 100 µm. **d** G′ and G″ of DNAgel on strain amplitude sweep. **e** ATR-FTIR analysis of pure salmon sperm DNA and DNAgel. **f** Images of water droplet absorbed by DNAgel. Scale bar: 1 mm. **g** Representative photographs of DNAgel in 2D and 3D shape during swelling process. Scale bar: 2 cm. **h** Mask Cutting, **i** macromolding for macro manufacturing (Scale bar: 2 cm) and **j** micromolding for micro-manufacturing (Scale bar: 2 mm) to obtain DNAgel in diverse shapes. Representative images are shown from three (**b**, **c**, **f**–**j**) independent experiments with similar results. Source data are provided as a Source Data file.

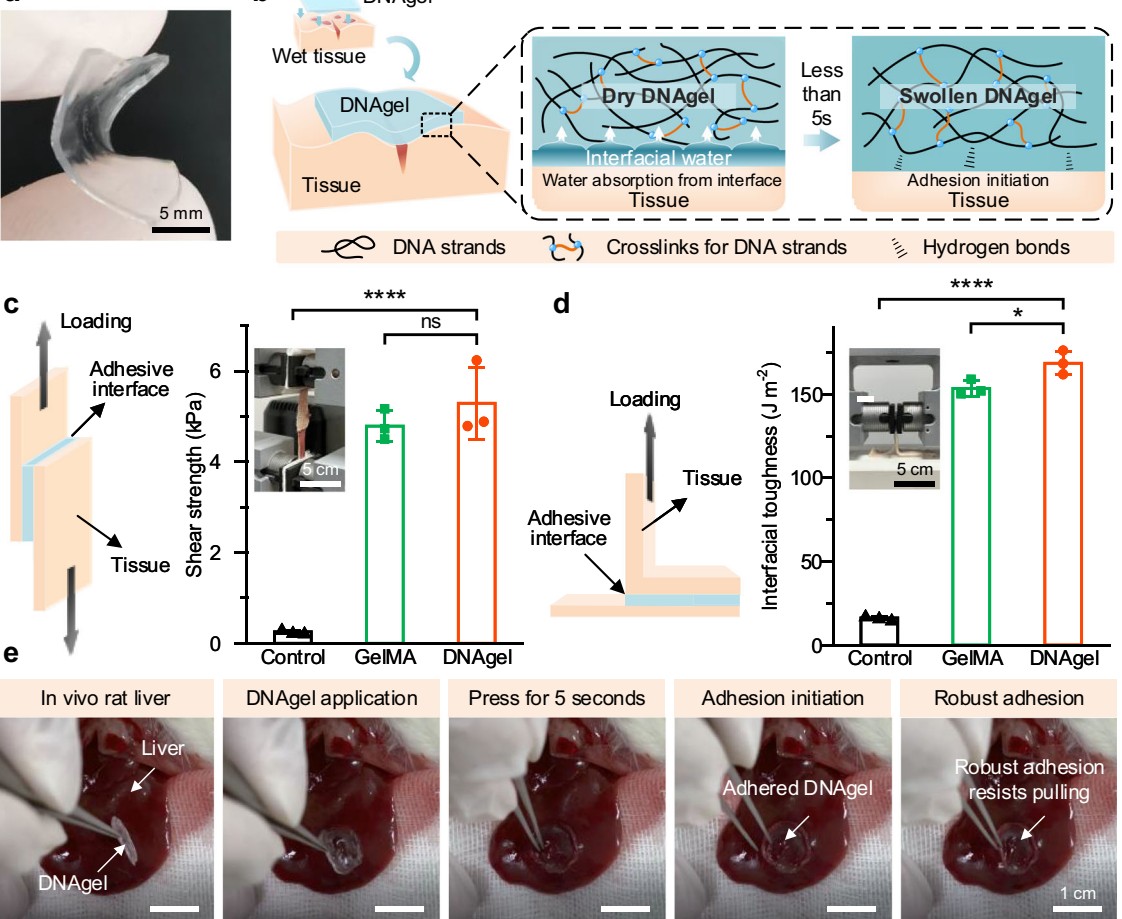

**Fig. 3 | DNAgel exhibited excellent wet adhesion to tissue. a** Bending shape of the DNAgel. Scale bar: 5 mm. **b** Schematic of adhesion formation between DNAgel and wet tissue. **c** Lap shear tests and **d** 90-degree peel tests of DNAgel with rat dorsal skin (1 cm*2 cm). Scale bar: 5 cm. Data were presented as mean ± SD ($n = 3$). **e** Adhesion of a DNAgel on rat liver in vivo. Scale bar: 1 cm. Statistical analysis was performed by one-way ANOVA with Tukey's multiple comparisons (**c**, **d**). Representative images are shown from three (**a**, **c**–**e**) independent experiments with similar results. Source data are provided as a Source Data file.

which can stick to the surrounding tissues of the bleeding point and perform as a physical barrier to promote hemostasis. Specifically, we employed DNAgel for hemostasis assays in three bleeding models, including rat-tail amputation, femoral artery injury and liver puncture wound models (Fig. 5). DNAgel exhibited an evident hemostatic effect in all models. For tail amputation model, DNAgel was applied and adhered to the section of bleeding tail, preventing the blood from flowing out of the wound. It was seen from Fig. 5a that the tail amputation model treated with DNAgel bleed less than the control and GS groups. The quantitative results further showed that the amount of blood loss in the DNAgel group was 0.04 ± 0.02 g, which was significantly reduced compared to that of the GS (0.36 ± 0.07 g) and control group (0.97 ± 0.3 g). It was indicated that the DNAgel could reduce the blood loss by 87.7% compared to GS. Then a rat femoral artery injury model was used to evaluate the hemostatic property of DNAgel on massive hemorrhage. Blood welled out from the wound when hemostatic forceps were opened, and three pieces of Gauze were completed wetted with blood in the control group. In addition, in the GS group, GS was immediately applied, followed by finger pressing for 30 s, when the femoral artery was ruptured. However, blood gushed out when the hemostatic forceps were removed. In comparison, the DNAgel still adhered and wrapped around the femoral artery injury after removing the external force, preventing the spillage of blood and promoting coagulation. The amount of blood loss in the DNAgel group was 0.33 ± 0.11 g, which was impressively less than the GS and control group. Furthermore, the

hemostatic capacity of DNAgel was evaluated in the liver model, which contains the most abundant blood vessels among visceral organs. Blood gushed out once the wound was created, and impregnated the entire gauze in the control group. In contrast, the areas of blood staining were smaller in the GS and DNAgel groups. In addition, the blood loss was in the following order: control group (5.87 ± 0.36 g) > GS group (2.07 ± 0.47 g) > DNAgel group (0.66 ± 0.32 g). The peritoneal lavage fluid samples were collected from rat femoral artery and liver puncture wound models, respectively. The DNAgel group showed the lowest relative O.D. value, indicating less blood loss and faster hemostasis capability for DNAgel (Fig. 5d).

## Mechanism of hemostasis via DNAgel

In order to further explore the molecular biological processes of DNAgel network in swelling, enrichment and activation of platelets and finally hemostasis, we analyzed the key molecules of platelet activation and clotting-related signaling pathways by transcriptome sequencing and flow cytometry. The proportion of activated platelets (CD62P positive cells) after DNAgel treatment was quantified by flow cytometry, while the platelet-rich plasma and/or whole blood treated with the TRAP-6, GS, DNA aq and no treatment were used as controls. The gating strategy was provided in Supplementary Fig. 11. It was shown in Fig. 6b, c that 85.26 ± 0.32% of platelets in the platelet-rich plasma of DNAgel group were activated, which was second only to the TRAP-6 positive control (90.23 ± 0.29%), and higher than the group of GS

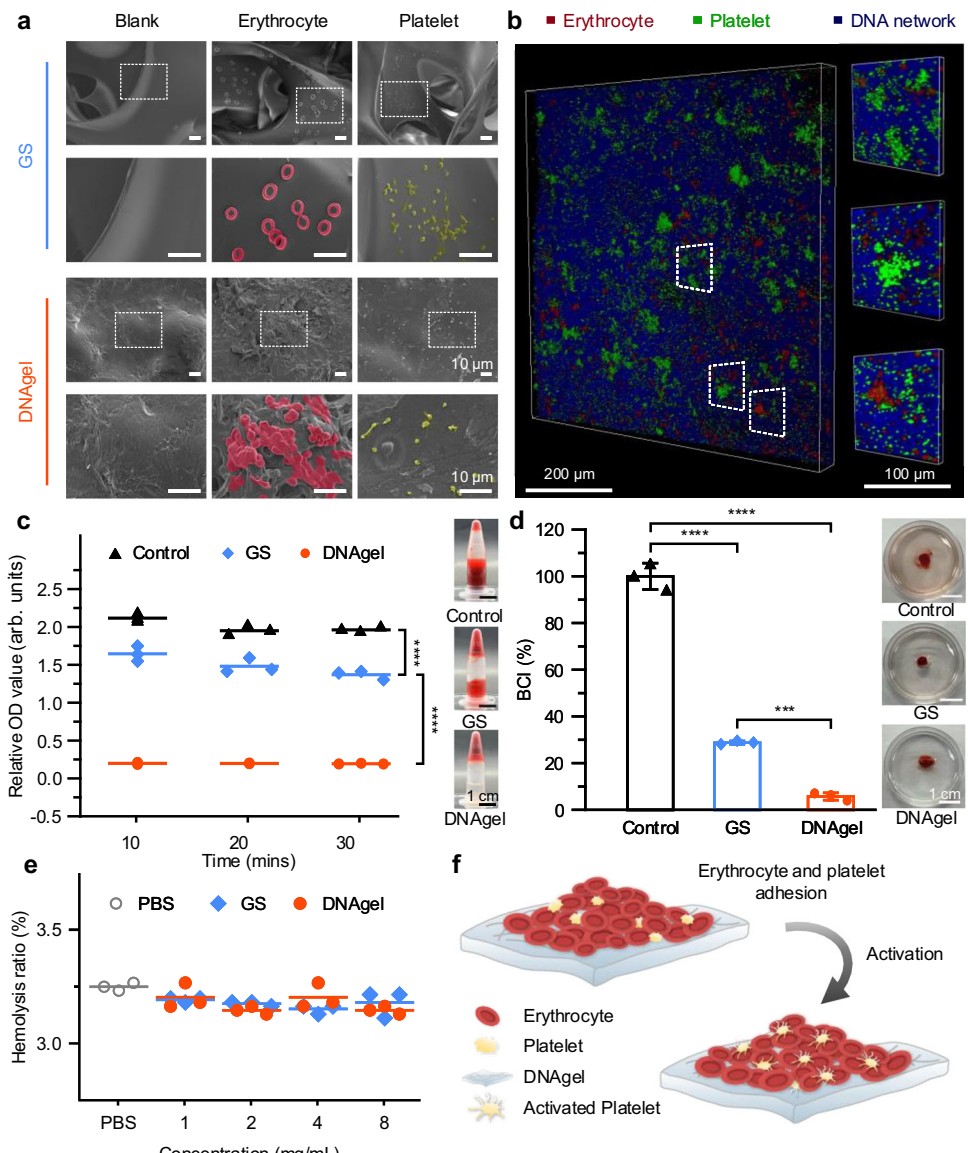

**Fig. 4 | DNAgel exhibited excellent in vitro procoagulant ability and biocompatibility. a** Representative SEM images of erythrocytes (red) and platelet (yellow) adhesion to the gelatin sponge (GS) with porous cavity and DNAgel with dense surface. Scale bar: 10 μm. **b** CLSM images of erythrocytes (red) and platelets (green) aggerated and adhered to the surface of DNA network (blue). Scale bar: 200 μm (left); Scale bar: 100 μm (right, enlarged). **c** Absorbance of blood clotting samples at different time points measured at 540 nm with UV–vis spectrophotometer and the representative pictures at 30 min. Scale bar: 1 cm. **d** Blood clotting index (BCI) of DNAgel and representative pictures, according to BCI(%) = $\frac{A_{\text{Experimental}}}{A_{\text{Negative}}} \times 100\%$. Scale bar: 1 cm. Data were presented as mean ± SD ($n = 3$ biologically independent samples). **e** Hemolysis ratio of DNAgel, according to Hemolysis ratio (%) = $\frac{A_p}{A_t} \times 100\%$. **f** Schematic illustration of adhesion and activation of erythrocytes and platelets on DNA gel. Statistical analysis was performed by one-way ANOVA with Tukey's multiple comparisons (**c**, **d**). Representative images are shown from three (**a**–**d**) independent experiments with similar results. Source data are provided as a Source Data file.

(78.73 ± 3.81%), DNAaq (66.73 ± 1.36%) and untreated (4.93 ± 3.01%) groups. Moreover, 71.23 ± 0.81% of the platelets in the whole blood of DNAgel group were activated, comparable to the TRAP-6 group (78.73 ± 0.44%) and apparently higher than the other three groups. Besides, the conformational change of platelet Glycoprotein IIb/IIIa (GPIIb/IIIa) was further investigated to characterize the activated status of platelets. It was shown in Supplementary Fig. 12 that 11.20 ± 0.17% of the platelets were PAC-1 positive in the DNAgel group, which was lower than that of the TRAP-1 group (34.17 ± 1.74%), but significantly higher than that of the GS group (5.05 ± 0.25%), DNAaq group (1.43 ± 0.09%) and untreated group (0.97 ± 0.14%). These results indicated that the DNA hydrogel we prepared could induce a conformational change of GPIIb/IIIa on the surface of platelets from the low-affinity state into the high-affinity state, thereby promoting

platelet activation and aggregation. To better understand the molecular mechanism of exogenous DNA materials as a biofunctional scaffold for platelet enrichment and activation in hemostasis at early stage of trauma, RNA sequencing transcriptome analysis was conducted on blood samples comparing the group of DNAgel with GS. It was observed that the expressions of hemostatic-related genes, including F5, PLCG2, Vav2, Vav3 and Dgka, were significantly upregulated in the DNAgel group after the initial coagulation signal was triggered compared to the GS group (Fig. 6d, e). The results detected by quantitative polymerase chain reaction (qPCR) and western blot also revealed the increased expression of PLCG2, Vav2 and Vav3 in the platelets treated with DNAgel (Supplementary Fig. 13). The DNA sequences of primers used for qPCR were provided in Supplementary Table 1. The content of activated coagulation factor VII (FVIIa) and

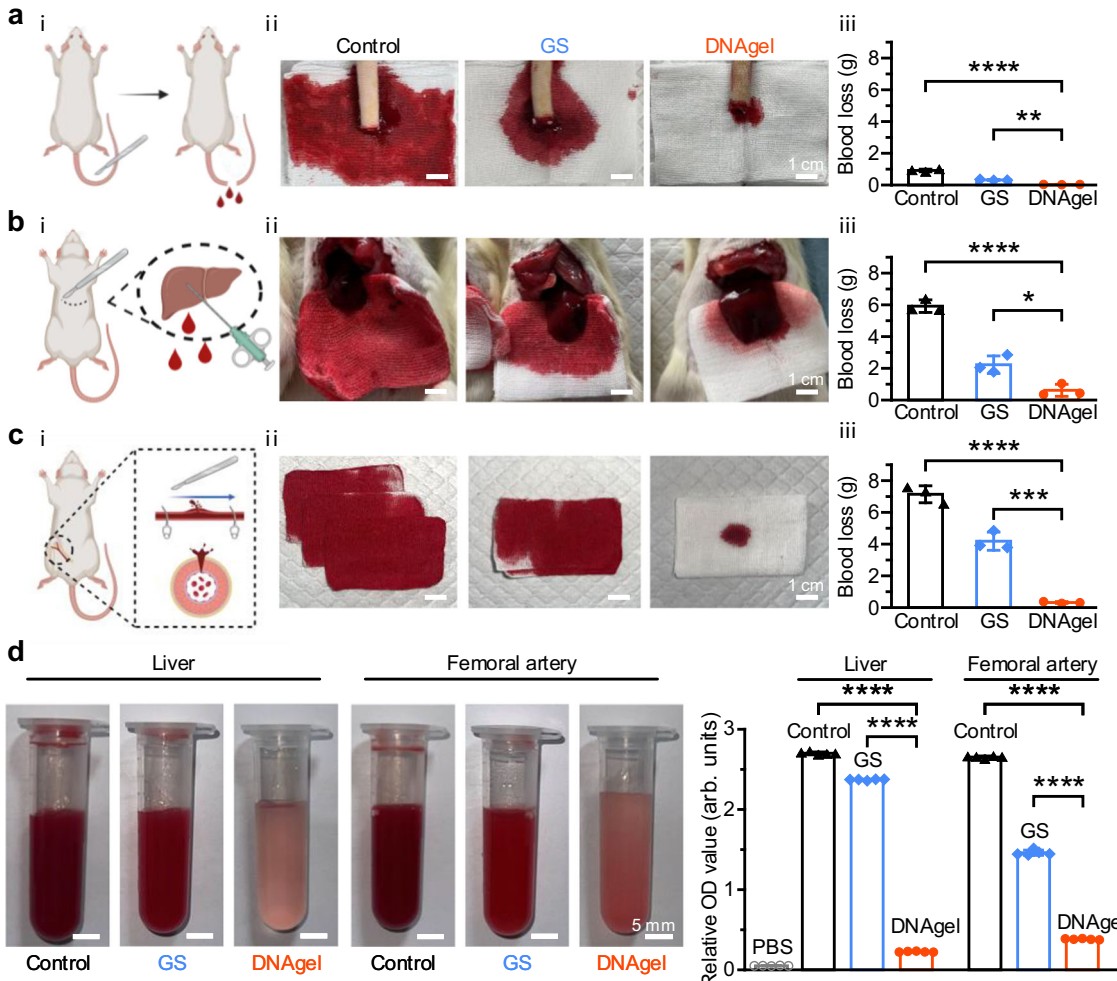

**Fig. 5 | DNAgel showed hemostatic performance in both superficial and deep bleeding models in vivo. a–c** Established SD rat tail amputation, liver puncture and femoral artery injury model, and the dynamic hemostatic ability using either gelatin sponge (GS) or DNAgel. Scale bar: 1 cm. Data were presented as mean ± SD (*n* = 3). Panels **a–c** was in part created with Biorender.com. **d** Peritoneal lavage fluid samples after femoral artery and liver hemostasis. Scale bar: 5 mm. Error bar, SDs (*n* = 5). (**p* < 0.05, ** *p* < 0.01, *** *p* < 0.001, and **** *p* < 0.0001). Statistical analysis was performed by one-way ANOVA with Tukey's multiple comparisons (**a–d**). Representative images are shown from three (**a–d**) independent experiments with similar results. Source data are provided as a Source Data file.

thrombin in the DNAgel-treated plasma were both increased, compared to the untreated plasma (Supplementary Figs. 14, 15). In addition, according to the GSEA analysis, the platelet formation, aggregation and activation pathways were significantly upregulated in the DNAgel group compared to the GS group (Fig. 6f). According to the above results, we visually demonstrated the molecular biological process of coagulation promotion by DNAgel in Fig. 6a. When contacted with DNAgel, the negatively charged surface of DNAgel may activate FSAP, which further activated the coagulation Factor VII in the blood, and enriched platelets. Once the initial coagulation signal was triggered, the expression of hemostatic-related genes was significantly upregulated, which further converted fibrinogen into fibrin and bound to GPIIb/IIIa on the surface of the platelet membrane, stimulating SFK activation[36,37]. Intracellular signaling may be transmitted by two pathways: i. SFKs activated SYK and promoted the assembly of the SLP76/LAT/Btk/Vav complex, which mediated PLCγ2 activation in a manner similar to the GPVI-mediated ITAM signaling pathway; ii. SFKs phosphorylated c-Src and activated RhoA GTP to inactivate RhoA, activating the signals associated with platelet activation[38]. Once platelets were activated, α-granules, dense granules and lysosomes were further released into the microenvironment[39], triggering a widespread coagulation cascade and promoting clot formation. It can be inferred from these results suggested that in addition to intercepting and enriching platelets and erythrocytes, the DNAgel network could further accelerate the maturation of blood clots through the FVII signaling pathway.

## Wound healing assessment of DNAgel and histological evaluation

Due to the effective wound-sealing property and accelerated hemostatic ability, DNAgel was further served as wound dressing for full-thickness skin defect repair. The degradability and therapeutic efficacy of DNAgel were monitored and photographed over the course of 0, 3, 5, 7 and 10 days and the quantitative wound areas and healing rates were also analyzed. The wounds treated with the GS and without treatment were set as control. As shown in Fig. 7, the DNAgel was degraded and disappeared within 2–3 days and the wound treated with DNAgel exhibited faster healing ability compared with the other two groups. The wound area in the DNAgel group contracted from 91.71 ± 1.50 mm$^2$ to 45.06 ± 4.37 mm$^2$ at Day 3, and the healing rate reached 50.91 ± 4.00%. The wound healing rate of DNAgel group continued to be faster than of the other two groups in the following days and became nearly 99.33 ± 0.14% at Day 10 (Fig. 7a–d). It was indicated that DNAgel had a better healing effect than the GS and control groups by comparing the wound contraction and healing rate. The detailed effects of DNAgel on wound healing at the tissue level were further assessed by histological analysis (Fig. 7e, f). The epidermal

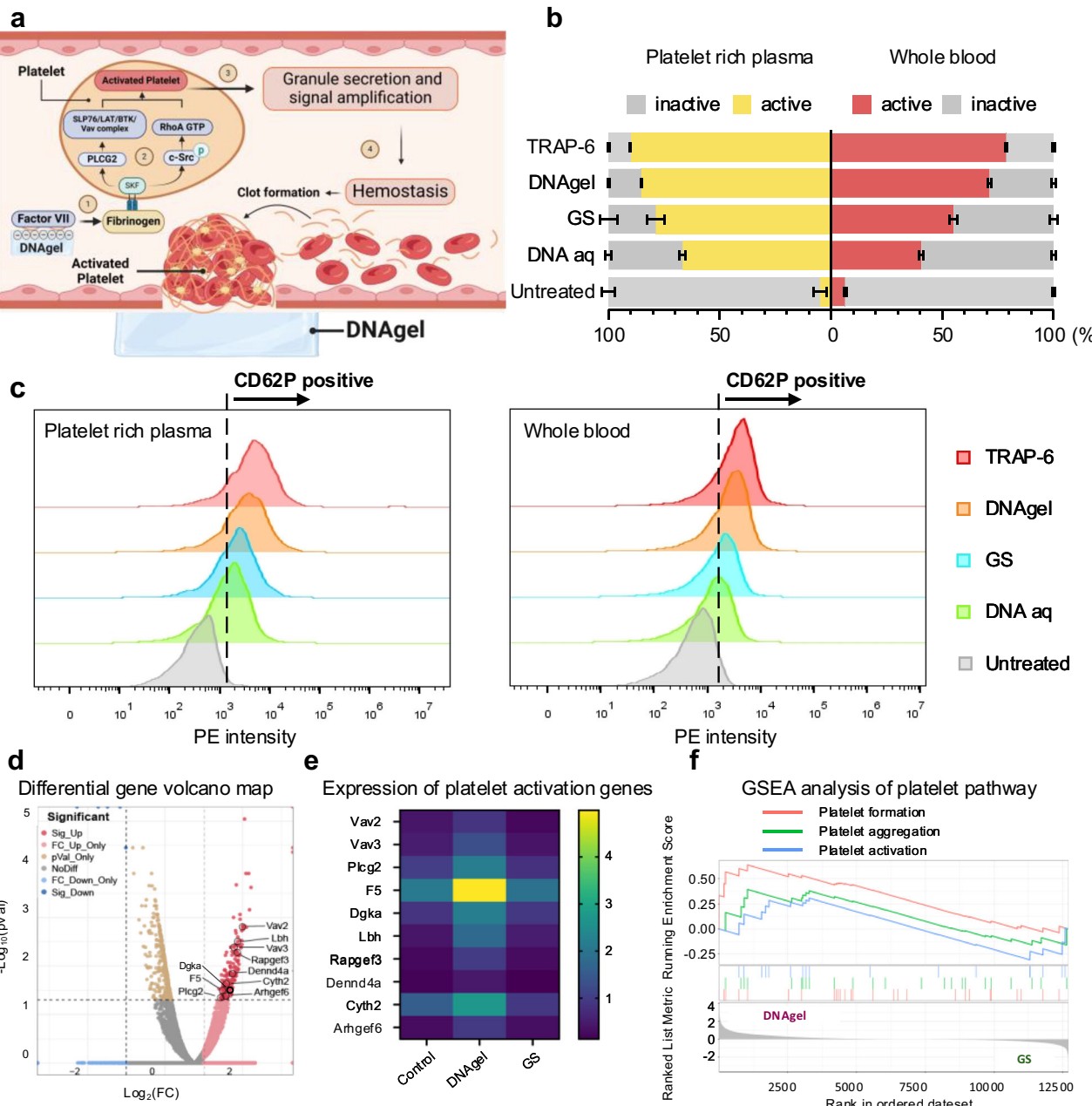

**Fig. 6 | DNAgel promoted platelet activation and accelerated the clotting process by upregulating related genes. a** Signaling pathway of clotting cascade triggered by DNAgel. Panel **a** was in part created with Biorender.com. **b** Statistical results of platelet activation and **c** representative flow cytometry plots under different treatment conditions. Data were presented as mean ± SD ($n = 3$). **d** RNA-seq analysis of whole blood cells, and differential expressed genes presented in a volcano map. Cutoff line: Log$_2$ fold change (Log$_2$FC) > |2| and adjusted $p$ value < 0.01. **e** Heatmap of differential expressed genes related to platelet activation and **f** GSEA of KEGG pathway analysis related to platelet. Statistical analysis was performed by one-way ANOVA with Tukey's multiple comparisons (**b**). Source data are provided as a Source Data file. RNAseq data generated in this study have been deposited in the GEO DataSets under accession code GSE268849.

renascence and inflammation in the wound area were evaluated by immunohistochemistry staining with cytokeratin 14 (CK14) on Day 7 and Day 10. As shown in Fig. 7e and Supplementary Fig. 14, the renascent epidermal thickness became thinner for all three groups during the healing process. There was no significant difference in epidermal thickness between the control and GS group on Day 10, while the DNAgel group exhibited a thinner epidermal similar to that of normal tissue, suggesting that DNAgel promoted the wound healing. It could also be observed from the Hematoxylin and Eosin (H&E) staining that a wide range of inflammatory cells were observed at Day 7 in the control and GS groups, indicating that severe inflammation remained uncontrolled. In addition, there was little new epidermal tissue in both

groups. However, only a few neutrophils appeared in the DNAgel-treated wound by Day 7. Notably, new blood vessels, renascent epidermis and hair follicles were observed, and basement membrane collagen formed in DNAgel group. At Day 10, the defect skin in the DNAgel group healed completely and was almost identical to normal skin, indicating dermal tissue regeneration, while the wounds remained opened and there were many inflammatory cells in the other two groups (Supplementary Fig. 16). The Masson's Trichrome staining and Sirius Red staining also showed that more regenerated collagen was deposited in the DNAgel-treated wounds when compared to the control and GS groups. Moreover, the quantification demonstrated that the wound tissue of DNAgel exhibited shorter wound lengths,

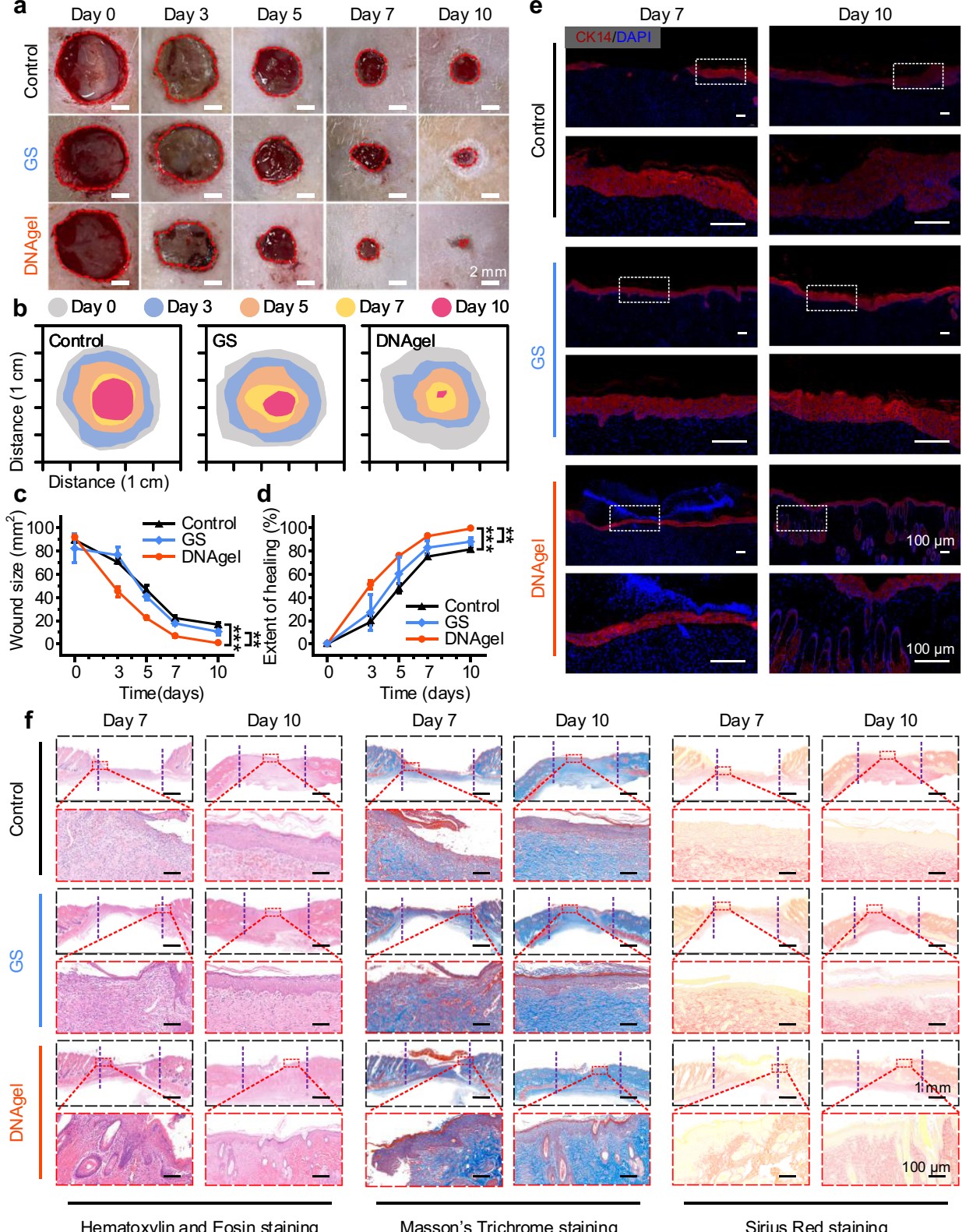

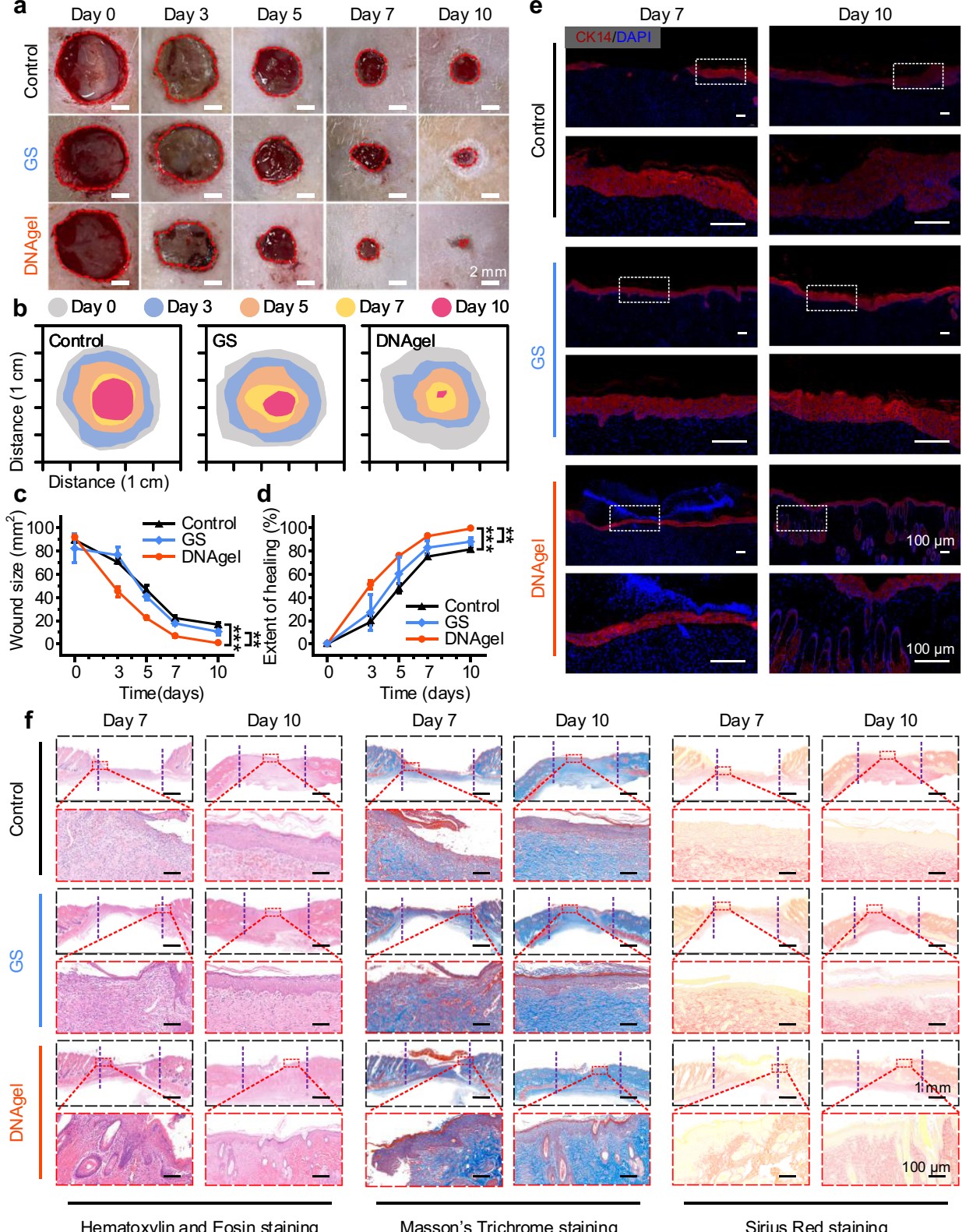

Hematoxylin and Eosin staining          Masson's Trichrome staining          Sirius Red staining

fewer inflammatory cells, more collagen-occupied area and more hair follicles, revealing the effects of DNAgel on accelerating wound healing. Besides, the H&E staining showed that the major organs of DNAgel group showed no significant difference from that of control group, indicating the excellent biocompatibility of DNAgel (Supplementary Fig. 17). The comprehensive analysis showed that DNAgel could accelerate wound healing, promote collagen formation and

regeneration of all layers of skin, including hair follicles and sebaceous glands.

## Discussion

In this section, we discuss the advantages and mechanisms of DNA hydrogel band-aid for accelerating hemostasis and wound healing. Previous studies have suggested that the emergency intervention such

**Fig. 7 | DNAgel promoted wound healing and completed skin regeneration.** Six full-thickness skin defects, 1 cm in diameter, were created symmetrically along the spine on the back of each rat using a hole punch and were treated according to the group. **a** Representative wound images during healing process on Day 0, 3, 5, 7, 10. Scale bar: 2 mm. **b** Wound traces at different days. **c** Wound size and **d** healing rate at different days for each group, according to Extent of healing (%) = $\left(1 - \frac{A_t}{A_0}\right) \times 100\%$. Data were presented as mean ± SD ($n = 3$). **e** Representative images of immunohistochemistry staining with CK14 (red) and DAPI (blue) on Day 7 (Wound closure time points in DNAgel group) and Day 10 (Total skin regeneration

time point in DNAgel group) for each group. Scale bar: 100 μm. **f** Hematoxylin and Eosin staining, Masson's Trichrome staining and Sirius Red staining on Day 7 and 10 of the newly regenerated skin tissues for each group. The purple dotted line marks the initial trauma margin. Scale bar: 1 mm (top); Scale bar: 100 μm (bottom, enlarged). Statistical analysis was performed by one-way ANOVA with Tukey's multiple comparisons (**c**, **d**). Representative images are shown from two (**a**, **e**, **f**) independent experiments with similar results. Source data are provided as a Source Data file.

as rapid sealing is critical in bleeding tissues[3,40,41]. DNAgel proposed in this study, with wide sources and easy accessibility, exhibits strong and stable adhesion to wet/bloody tissues and serves as a physical barrier against bleeding. When applied to the trauma, the highly hydrophilic DNAgel with rough surface and porous structure rapidly absorbs water from the environment (Fig. 2c, f), which causing the distance between the DNAgel and the tissue to decrease. In contrast, GS is not good at blood absorption and tissue adhesion due to the poor hydrophilicity (Supplementary Fig. 18). Subsequently, the DNAgel immediately forms intermolecular bonds with the tissue surfaces through hydrogen bonding and electrostatic interaction, resulting in stable adhesion. Compared to hydrogels that crosslinked in situ, the pre-crosslinked DNAgel enables fast response without any mixing or handling, and avoid potential mutagenic and cytotoxic DNA lesions caused by UV irradiation[3,4,42,43]. Besides, the swollen DNAgel demonstrates the potential treatment for noncompressible hemorrhage (Figs. 2g and 5b). When stuffed into the liver wound, the DNAgel is supposed to rapidly absorb blood and expand to fill the entire lesion, similar to the self-expanding polyurethane polymer[44–46], and causes tissue compression in confined spaces of the body[47], forcing the blood vessels to constrict and slowing the blood flow[13,48]. The application of DNAgel to the bleeding wound can also avoid tissue damage when compared to zeolite-based hemostatic agents, which would generate heat when absorbing blood. The stable adhesion and rapid hemostasis promote the formation of a physical barrier and prevent bacterial infection. Furthermore, DNAgel facilitates the aggregation of blood cells and the absorption of blood plasma. In this study, we find that red blood cells and platelets could aggregate and attach to the surface of the DNAgel (Fig. 4a, b), providing a scaffold for platelet activation and promote thrombosis, similar to the natural neutrophil-derived extracellular DNA for physiological clotting according to the previous study[31]. According to the flow cytometry results, the proportion of platelet activation in the DNAgel group was higher than that in the DNA aq group, especially in the whole blood samples, indicating that the absorption of fluid from the blood by DNAgel, in the presence of RBCs (Fig. 6b, c), may further trigger the coagulation cascade, promoting platelet activation and hemostasis. On the contrary, the pores of nanogels may be too small for cells to pass through and cause RBCs to separate from the plasma rich in platelets and clotting factors, thus slowing down the clotting process.

More importantly, DNAgel may play a key role in the later wound healing process. The stable adhesion allows DNAgel to bring the cut sides together, like a suture, thereby promoting healing. Meanwhile, the high-swelling and porous properties suggest that the DNAgel could behave like a bridge for cell migration, and allows the two sides of wound to meet together and eventually heal. It is indicated that fibrosis may be reduced due to the expansion of gel and temporary space occupation. The disappearance of DNAgel in Fig. 7 and Supplementary Fig. 3 suggests that the long strands of DNA are biodegraded into nucleotides and then assimilated by the quick dividing cells which involved in the healing process. Another benefit is that DNAgel does not require a second extraction after a blood clot has formed, especially for deep wounds, which greatly reducing the potential risk of infection and secondary damage. In the wound healing experiment, we found that the DNAgel group could promote the complete healing of full-thickness skin defects when compared with the control group and

the GS group, the healed tissue of the DNAgel group showed intact hair follicles and sebaceous glands. This may be related to the activation of innate immune pathways and the promotion of tissue regeneration by DNAgel, which requires further investigation.

In brief, we proposed a DNAgel band-aid for emergency hemostasis and accelerated wound healing. The as-prepared hydrogel shows swelling ability and instant and robust tissue adhesion to cope with the emergency of rapid bleeding in the deep. We further validated the feasibility and efficacy of DNAgel band-aid as a hemostatic agent. The DNAgel has a low hemolysis and good aggregation ability of blood red cells and platelets. The blood loss of the DNAgel groups in different hemorrhage models are significantly lower than those of the control groups, indicating that DNAgel has excellent in vivo hemostatic properties as a sealant and stuffing. Furthermore, the DNAgel band-aid possessed a better wound-healing effect than the GS. Overall, DNAgel is an effective candidate for future clinical use in deep bleeding control and emergency wound care.

## Methods
### Materials
Salmon Sperm DNA (highly molecule weight, linear), poly (ethylene glycol) diacrylate (PEGDA, average Mn 575), and sodium hydroxide (NaOH, 96%) were purchased from Coolaber, Sigma-Aldrich and Aladdin, respectively. Gelatin sponge (GS) was purchased from Hushida Medical. Sodium chloride and ethanol absolute were purchased from Sinopharm. Normal saline and phosphate-buffered saline (PBS) were bought from Cisen and Basal Media, respectively. ACD anticoagulant (r10201) was purchased from Leagene Biotechnology, $K_2$EDTA blood collection tuber (367863) was purchased from Becton Dickinson, and platelet separation medium was purchased from Solarbio. Cell Counting Kit-8 (CCK-8) and Calcein/PI Cell Viability/Cytotoxicity Assay Kit were bought from Med Chem Express (MCE) and Beyotime, respectively. Human Activated Coagulation Factor VIIa (FVIIa) ELISA Kit, and Rat Thrombin ELISA Kit were purchased from MeiaoBio and Bioswamp, respectively. PLCG2 Monoclonal antibody, VAV2 Polyclonal antibody, VAV3 Polyclonal antibody, and GAPDH Monoclonal antibody were purchased from Proteintech. Phospho-PLCγ2 (Tyr759) (E9E9Y) Rabbit mAb and Multicolor Prestained Protein Ladder (WJ103) were bought from Cell Signaling Technology and Epizyme, respectively. Antibodies of FITC Mouse Anti-Rat CD61, PE Mouse Anti-Rat CD62P, and PE Anti-Human CD61 were purchased from Biolegend and FITC Anti-Human PAC-1 was purchased from Thermo-Fisher. Deionized water (DW) used in this study was produced by Pall Cascada™ lab water purification system. Silicon rubber molds were customized from Wenext. Sprague-Dawley (SD) rats between 200 and 250 g of weight, at 6 weeks of age, and C57BL/6 mice at 6 weeks of age were purchased from SLAC Animal.

### Preparation of DNAgel
Salmon testes DNA strands were covalently crosslinked by poly (ethylene glycol) diacrylate (PEGDA) to form 3D hydrogel networks. Specifically, the DNAgel precursor was prepared by dissolving 6 wt% salmon tests DNA in NaOH (0.127 M) solution at room temperature. The crosslinker, PEGDA (Mn = 575), was added at a mass/volume ratio of 2:1 to DNA, and then mixed thoroughly and rapidly with the

hydrogel precursor. The reaction solution was centrifuged $2000 \times g$ for around 20 s to eliminate air bubbles and then transferred into the custom molds. To speed up the reaction, the mold containing the DNA solution was placed on a heating-stage at 37 °C for 1 h to complete the gelation. The hydrogels were rinsed with DW and PBS to remove untreated chemicals, dried on the 37 °C-heating stage and stored sealed at room temperature for later use.

## Physical and chemical characterization

The morphology of DNAgel were evaluated using an FE-SEM, JSM-IT800 (JEOL, Japan). The freeze-dried swollen DNAgel sample was sputter coated with gold and observed at an acceleration voltage of 3 kV. The rheological analyses were performed in a HAAKE MARS 40 Rheometer (Thermo Scientific, German), and tested by using an 8 mm steel plate. Oscillatory strain amplitude sweeps were conducted at a frequency of 1 Hz at 25 °C. Oscillatory frequency sweep measurements were conducted at a certain strain amplitude. All rheological experiments were performed in triplicate, and storage modulus (G′) and loss modulus (G″) were recorded. A Nicolet FTIR spectrophotometer (Thermo Scientific, German) was used to record the FTIR spectra of the freeze-dried samples, in the range of 400 up to 4000 cm$^{-1}$.

## Swelling capacity of DNAgel

The swelling ratio (SW) was used to evaluate the swelling capacity of DNAgel. A dried DNAgel was soaked in DW and then the hydrogel was removed at selected time intervals and weighed after removing the excess water from the gel surface with a filter paper. The The SW was calculated according to the following equation: $SW = W_t / W_0$, where $W_t$ and $W_0$ denote the weight of the swollen hydrogel and the initial weight of the dried hydrogel, respectively. The SW of DNAgel swelling in PBS and heparinized fresh whole blood were also measured by the same method as above.

**Tissue adhesive properties.** The tissue adhesion properties of DNAgel were evaluated by lap shear test and 90° peeling-off, respectively, using a universal testing machine (CMT6103, MTS, USA). Fresh rat dorsal skin was cut into pieces of 60 mm × 20 mm × 1 mm. The DNAgel was attached directly between the surfaces of two pieces of rat skin, maintaining the overlapping area at 20 mm × 2 mm. The sample was then kept at ambient temperature for 20 min, followed by mechanical measurement of shear strength and interfacial toughness by the machine. The tests were performed in triplicate at a speed of 10 mm/min.

**Erythrocyte and platelet adhesion.** Heparinized whole blood was collected from healthy rats and centrifuged at $200 \times g$ for 20 min at room temperature to obtain platelet-rich plasma (PRP). 50 μL of whole blood or PRP was added to the dried DNAgel. After incubation at 37 °C for 30 min, the hydrogel was rinsed with PBS to remove non-adherent blood cells or platelets. It was then fixed with 4% paraformaldehyde for 2 h, followed by sequential dehydration with 25, 50, 75, 85, 90, and 100% ethanol solutions for 15 min, respectively. The dried sample was attached to a metal stage with double-sided adhesive carbon tape, sputter-coated with a thin layer of gold film and then observed by SEM at an accelerated voltage of 5 kV. Commercial GS was used as a control. For the immunofluorescence, the DNAgel was immersed in fresh anticoagulated whole blood until the DNAgel was fully swollen. It was then washed in PBS buffer containing 1% BSA to remove blood cells that did not adhere to the DNAgel. The DNAgel was then fixed in 4% paraformaldehyde. Mouse Anti-Human CD61 (BD Biosciences) was used as primary antibody to stain platelets, and accordingly FITC Conjugated Rabbit Anti-Mouse IgG Rabbit Polyclonal Antibody (Huabio) was used as secondary antibody. After platelet staining, Hoechst 33342 (Thermo fisher) was used to stain the DNA. Due to the spontaneous red fluorescence of its hemoglobin, red blood cells can be directly observed under RFP excitation light without staining.

## BCI test

The BCI was used for a preliminary assessment of the hemostatic capacity of DNAgel. The DNA gel cut into cylinder (diameter: 8 mm, height: 5 mm) was added to the culture dish, and preheated in a 37 °C for 5 min. The 50 μL anticoagulated whole blood collected from healthy rats was then dropped onto the DNAgel and incubated in the 37 °C for another 5 min, followed by 10 μL of CaCl$_2$ (0.2 M) solution. After 5 min, 10 mL of DW was slowly added into the dish without disturbing the coagulated blood and incubated at 37 °C for another 5 min to collect the free RBCs that were not trapped in the blood clot. Commercial GS with the same volume and group without adding any materials were set as controls. The absorbance of the hemoglobin solution for each sample was determined at 540 nm with a UV–vis spectrophotometer. The blood clotting index (BCI) was expressed as follows: BCI (%) $= \frac{A_{\text{Experimental}}}{A_{\text{Negative}}} \times 100\%$, where $A_{\text{Experimental}}$ represents the absorbance of the hemoglobin solution in the experimental groups and $A_{\text{Negative}}$ represents the absorbance of the hemoglobin solution diluent without any other treatment in the negative control groups.

## In vitro hemostatic performance

Cylindrical DNA gel (diameter: 8 mm, height: 5 mm) was first added into the 1.5 mL centrifuge tube and preheated in a 37 °C water bath for 5 min. GS with the same volume and tube without any materials were set as the controls. 500 μL anticoagulated whole blood was added and incubated in the 37 °C for another 5 min, followed by 50 μL of CaCl$_2$ (25 mM) solution. The tubes were tilted every 15–30 s and photographed to determine whether the blood was coagulated. Additionally, 500 μL of DW was slowly added into the tubes at 10, 20, 30 min, respectively, and incubated at 37 °C for another 5 min, and the absorbance of the supernatant for each sample was determined at 540 nm with a UV–vis spectrophotometer subsequently.

## Biocompatibility tests

In vitro cytocompatibility tests were evaluated by culturing human normal hepatocyte line WRL68 cells with DNAgel-conditioned medium. To obtain DNAgel-conditioned medium, 20 mg of DNAgel was first incubated in 1 mL of Dulbecco's modified Eagle medium (DMEM) at 37 °C for 24 h. Pristine DMEM was used as a control. WRL68 cells were plated in 96-well plates with a density of $1.5 \times 10^5$ cells/mL, treated with the conditioned medium, and subsequently incubated at 37 °C in 5% CO$_2$ for 24 h or 48 h. Cell viability was determined by a live-dead fluorescence staining assay, and observed by confocal laser scanning microscope (A1 HD25, Nikon, Japan). Furthermore, Cell Counting Kit-8 (CCK-8) assay was also conducted to quantify cell viability. The procedure for WRL68 cells culture was the same as described above. The optical density (OD) at 450 nm was measured using a microplate reader (TECAN, SPARK, Austria) at 24 and 48 h, respectively. And the cell viability was calculated according to the following equation: Cell viability $= \frac{OD_{DNAgel} - OD_{Blank}}{OD_{Control} - OD_{Blank}} \times 100\%$. Besides, in order to evaluate the acute toxicity of the DNAgel, 6-week-old C57BL/6 mice were anesthetized with isoflurane and a 5-mm-long cut was made on the dorsal skin of mice. 5 mg DNAgel was then implanted subcutaneously. The major organs were collected after 24 h and stained with H&E, including the lung, liver, spleen, kidney, and heart. The experiment was approved by the Experimental Animal Welfare & Ethical Review Committee of Zhejiang university (Protocol No. ZJU20210268) and carried out in strict accordance with guidelines for the protection and use of experiment animals.

## Hemolysis ratio of DNAgel

The hemolysis ratio was first measured to assess the blood compatibility of the DNAgel. Erythrocytes were separated from the rat blood by centrifugation at $116 \times g$ for 10 min. The erythrocytes obtained were washed with PBS for three times and then diluted to a final

concentration of 2% (v/v). The lyophilized DNAgel or GS was added in PBS at 1 mg/L, 2 mg/L, 4 mg/L, 8 mg/L to obtained sample solutions, respectively. The sample solution (500 μL) with the erythrocyte stock (500 μL) was gently mixed in EP tubes, then shaken at 37 °C for 1 h at a speed of 150 rpm. The tubes were then centrifuged at 116 × $g$ for 10 min and the absorbance of supernatant (100 μL) was read at 540 nm using a microplate reader. 0.1% Triton x-100 was used as a positive control while PBS was used as a negative control. The hemolysis percentage was calculated according to the equation: Hemolysis ratio (%) = $\frac{A_p}{A_t}$ × 100%, where $A_p$ was the absorbance value for the experimental group or PBS, $A_t$ was the absorbance value for the Triton X-100 positive control.

## In vivo hemostatic effect test in rats

All animal experiments were carried out in strict accordance with guidelines for the protection and use of experiment animals in Institute of Basic Medicine and Cancer, Chinese Academy of Sciences (IBMC). The experiments were approved by the Animal Care & Welfare Committee of IBMC (Protocol No. 2022R0015). Male Sprague-Dawley rats (200–250 g) were used to assess the hemostatic capacity of DNAgel. All the SD rats were anesthetized with isoflurane, and restrained onto a wooden corkboard for surgery. For tail hemostasis, a third of the tail was cut with surgical scissors while a pre-weighed gauze was placed beneath the wound, and then exposed to air for 15 s to allow free bleeding. The wound was then covered with a 1 cm diameter round DNAgel or GS, or left untreated. We defined the end point of hemostasis as the blood-stained area on the gauze didn't expand and no blood seeps out of the seam at DNAgel/GS-wound contact surface. The mass of blood absorbed on the gauze was recorded. For rat femoral artery hemostasis, skin and soft tissues were peeled off by the surgical scalpel to expose the femoral artery. The proximal and distal ends of the artery were clamped with hemostatic forceps and then incised with a bistoury. A piece of DNAgel was wrapped around the incision of the femoral artery, and then the clamps were released on both sides. Untreated group and 2 min-finger pressing group were used as controls. For hemostasis of the liver, the chest of the rat was opened to expose the liver, and a pre-weighted gauze was placed under the liver. A 0.5 × 0.5 × 0.5 cm³ wound was created in the liver by biopsy needle. After bleeding for 15 s, DNAgel was gently stuffed into the liver puncture wound. Blood loss was recorded ($n \geq 3$).

## Flow cytometry

5 mL of whole blood or PRP was first treated with dried DNAgel (20 mg) for 15 min, respectively. Cells were then stained with FITC Mouse Anti-Rat CD61 and PE Mouse Anti-Rat CD62P. Data were collected and analyzed on a flow cytometer (CytoFLEX LX, Beckman Coulter, USA). The PRP or/and whole blood treated with the TRAP-6 (2 μM), GS, DNA aq (6 mg/mL) and no treatment were set as control. In addition, the antibody against PAC-1 was further used to evaluate the conformational change of platelet Glycoprotein IIb/IIIa (GPIIb/IIIa), to characterize the activated status of platelets. Fresh human whole blood was used, and cells were stained with PE Anti-Human CD61 and FITC Anti-Human PAC-1. The subsequent steps were similar to above.

## Western blot

1 mL of whole blood treated with dried DNAgel (5 mg) for 15 min, and lysed with RIPA buffer on ice for 30 minutes. The supernatants of the cell lysate was isolated at 16,200 × $g$ for 15 min at 4 °C, then denatured at 95 °C for 10 min in 1×Loading buffer (10% glycerol, 50 mM Tris-HCl (pH 6.8), 2% β-Mercaptoethanol, 0.02% Bromophenol blue, 2% SDS). Denatured proteins and multicolored prestained protein ladder (Epizyme, WJ103) were separated by SDS-PAGE and then transferred to polyvinylidene difluoride (PVDF) membrane (Millipore, IPVH00010). After incubation with primary and horseradish peroxidase (HRP)-conjugated secondary antibodies, protein bands were detected by BeyoECL Moon Kit (Beyotime, P0018FM) and scanned with ChemiDoc Touch System (Bio-Rad USA). Untreated whole blood sample was set as control.

## RNA sequencing transcriptome analysis

The fresh whole blood of rats was collected and incubated with DNAgel and GS for 15 min, respectively. The high-throughput RNA sequencing (RNA-seq) and transcriptome analysis were performed by LC-BIO Biotech Ltd (Hangzhou, China). Bioinformatic analysis was performed using the OmicStudio tools at https://www.omicstudio.cn. The volcano plot, heatmap and GSEA plot were drawn based on the R version 4.1.3 (2022-03-10) on the OmicStudio platform.

## In vivo wound healing

The wound healing experiments were conducted by a full-thickness skin detect model of SD rats according to previous work. SD rats were firstly anesthetized with isoflurane, and the pelage on the back was shaved. The round injures with a diameter of 1 cm were created on the dorsum of rats by surgical scissors. Subsequently, the wounds were covered with DNAgel and GS in equal size, or without treatment, respectively. Digital photographs were taken at different time intervals (Day 0, 3, 5, 7 and 10) and the wound area was measured using Adobe Illustrator 2020 software. The healing rate (%) was calculated by the following equation: Extent of healing (%) = $(1 - \frac{A_t}{A_0})$ × 100%, where $A_0$ was the wound area of Day 0, $A_t$ was the wound area of Day $t$ ($t$ = 3,5,7, or 10).

## Histopathological study

The wound histology specimens were collected on Day 7 (wound closure time points in DNAgel group) and Day 10 (total skin regeneration time point in DNAgel group), and all samples were fixed in 4% paraformaldehyde solution for 12 h, followed by embedded in paraffin to prepare 5-μm-thickness tissue sections. Immunofluorescence staining (CK14) was conducted to evaluate the collagen deposition and angiogenesis. Besides, the specimens were strained with Hematoxylin and Eosin (H&E), Masson trichrome, and Sirius red, respectively, to assess the morphology, tissue regeneration and collagen deposition.

## Statistical analysis

Statistical analysis was performed by one-way ANOVA with Tukey's multiple comparisons or unpaired two-tailed Student's $t$ testing, using GraphPad Prism (v9.0). Generally, the differences were statistically significant at *$p < 0.05$, **$p < 0.01$, ***$p < 0.001$, and ****$p < 0.0001$. The presented data indicate the mean ± standard deviation (SD) from at least three independent experiments.

## Reporting summary

Further information on research design is available in the Nature Portfolio Reporting Summary linked to this article.

# Data availability

The authors declare that the data supporting the findings of this study are available within the paper and its supplementary information. Source data for each graph and uncropped blot images are provided as a Source Data file. Source data are provided with this paper.

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

## Acknowledgements

The authors are grateful for the financial support from the National Key R&D Program of China (No. 2022YFA1304500 and 2021YFF1200200, to J.S.), the National Natural Science Foundation of China (Nos. 22161132008, to J.S.; Nos. 8212200044 and 82071085, to M.Y.), the Starry Night Science Fund of Zhejiang University Shanghai Institute for Advanced Study (SN-ZJU-SIAS-006, to J.S.), and Zhejiang Provincial Natural Science Foundation of China (YXD23B0301, to J.S.; LR21H140001, to M.Y.). J.S. also acknowledges the support from Youth Cross-disciplinary Team Project of the Chinese Academy of Sciences and Leading Talents of Health Profession Training Project of Zhejiang Province. Some of the figures were created using Biorender.com.

## Author contributions

M.Y. and J.S. designed research; R.Y., Z.Z., T.G., D.C., K.J., and Q.D. performed research; R.Y., Z.Z., T.G., S.Z., K. X., and Y. J. analyzed data; and R.Y., Z.Z., P. C., and D. L., M.Y. and J.S. wrote the paper.

## Competing interests

The authors declare no competing interests.
