## [Peer Review File · Nature Communications]

Reviewers' Comments:

Reviewer #1:

Remarks to the Author:

In this study, the authors demonstrated rapid hemostasis and effective wound healing of DNA-based hydrogels, inspired by neutrophil extracellular traps (NETs), a mesh-type network of DNA and proteins released from cells. Although the experimental data on materials characterization, in vivo hemostasis, and wound healing capability support the authors' claims, the novelty and originality of the study are highly insufficient. Firstly, the authors utilized previously reported DNA-based hydrogels (J. Am. Chem. Soc. 2020, 142, 22, 10114), indicating a lack of novelty in the materials. Additionally, the hemostatic efficacy and coagulation mechanism of DNA-based materials (relevant to polyphosphate materials) have already been reported in other references (Adv. Funct. Mater. 2015, 25, 1270–1278; Macromol. Biosci. 2017, 17, 1600252; Green Chem., 2021, 23, 629–669; Res Pract Thromb Haemost. 2019; 3: 18–25). The reviewer thinks that the only difference in this study from others was a specific biological analysis regarding the signaling pathway of the clotting cascade (Figure 6), but that is not enough to be published in this journal. The manuscript should be transferred to other sister journals. There are specific comments for further improvement.

< Major issues >

1. Figure 3: The tissue-adhesive strength of DNAgel (~5 kPa) seems too low to overcome the large transformation during isotropic swelling. The authors should compare its strength to other adhesives, not gelatin sponge. The gelatin sponge cannot be a good control, and the authors should compare the adhesiveness and performance of DNAgel to that of clinically useful hemostatic agents, such as fibrin sealants.
2. Figure 4: The authors said that DNAgel could capture many erythrocytes and platelets in vitro. The experiments should be re-conducted for the sample after in vivo hemostasis tests. In addition, the pore size of the dried DNAgel should be quantitatively analyzed to investigate whether RBCs with a large size or platelets can be trapped in the gels by physical absorption or other chemical interactions (e.g., electrostatic forces).
3. Is there any significant difference in hemostatic performance between a dried bare DNA film (without PEGDA) and the DNAgel?
4. The authors emphasized the importance of the high liquid absorption capacity and the swelling behavior of DNAgel. Nonetheless, the investigation of hemostasis does not represent these characteristics. The authors should note the hemostatic effect due to blood absorption and swelling to emphasize the properties of the DNAgel. Alternatively, the high liquid absorption capability and linear swelling property of the DNAgel should be simply mentioned as properties of the DNAgel. Similarly, during the hemostasis process using DNAgel, more experiments are needed to show that swelling and expansion of the gel prevent blood outflow.
5. According to the results of flow cytometry, RNA sequencing, and GSEA analysis (Figure 6), platelet activation and coagulation acceleration effects of DNA gel were demonstrated. However, the molecular biological process of it should be explained more detailed. Stephanie et al. [<https://doi.org/10.1073/pnas.0507195103>] reported DNA failed to abrogate the tissue factor pathway inhibitor (TFPI) function, unlike polyphosphate molecules, and Figure 6B showed whole blood with DNA aq was not significantly different from the untreated group. Please explain in more detail "how the negatively charged surface of DNA gel affects the coagulation factor VII signaling pathway" with some results or references.
6. In the wound healing section, the enlarged position of the histology images is different for each group. Because there are differences in the progress of regeneration between the edges and the central lesion of the damaged area, the authors need to show images from the same location as much as possible.

< Minor issues >

1. The "Experimental Section" should be checked and explained in more detail. For example, the

authors do not provide the rheological measurement procedure (Figure 2D). Also, in other parts, the experimental procedure should be re-written to provide details.

2. Recent references regarding hemostasis of DNA-based materials should be cited (Adv. Funct. Mater. 2015, 25, 1270–1278; Macromol. Biosci. 2017, 17, 1600252).

3. Was the DNA gel used for investigating rheological properties (Figure 2D) measured after fully swelling? If not, it should be suggested to ensure in vivo stability.

4. Can rheological properties and swelling behaviors be changed due to swelling or adhesion of blood components (e.g., RBCs, platelets, plasma) instead of DW?

5. The authors mentioned that the commercially available absorbable gelatin sponge (GS) is not good at blood absorption in the Discussion section. To demonstrate it, its contact angle (Figure S6) at various time scales should be compared with that of DNA gel (Figure 2F).

6. Please check entire manuscript for typos.

For example,

Page 17, Lines 266-271 "pathways: i . SFKs activated SYK" should be "pathways: i . SFKs activated SYK".

Figure 7E, color labels were missing

...

Reviewer #2:

Remarks to the Author:

The authors aimed at developing a novel, DNAgel band-aid with strong hemostatic properties. The DNA hydrogel band-aid was adhesive, procoagulant, biocompatible, and promoted wound healing.

The biophysical properties of the DNA hydrogel band-aid were examined, showing in three rat bleeding models that it prevented blood loss. The DNA hydrogel band-aid was also biocompatible in in vitro cell culturing and accelerated wound healing in an in vivo skin incision model.

As a major drawback, the procoagulant activity was mainly mediated platelet activation - which was based on flow cytometric and RNA-Seq analysis neglecting plasmatic coagulation.

Major comments:

1) The flow cytometry study was not well performed, and data analysis is questionable:

Figure 6B: Untreated samples of PRP and whole blood show already 50% preactivated platelets (acceptable would be lower than about 10%)! In this regard, PRP generation should be performed at RT and not at 4 °C (stated in the Methods, page 18) as platelets are sensitive to cold and could get preactivated.

Figure 6C: x-axis and y-axis labeling of all plots (CD62P/CD61) and quadrant labeling below untreated is not correct (CD62P+ is determined as inactive platelets! CD62P is a platelet activation marker); according to the Methods, the x-axis should be CD61 (FITC Mouse Anti-Rat CD61), and y-axis should be CD62P. How was the gating strategy?

All platelets should have been CD61-FITC-positive as this is a platelet lineage marker (expressed on the surface without and with activation), but the representative plots provided in Figure 6C only show marginal FITC-positivity. Were the antibodies titrated for saturating concentrations? Were isotype control antibodies used to distinguish between specific and unspecific antibody binding?

The use of CaCl₂ might not be an ideal positive control for analyzing P-selectin expression as it activates the coagulation cascade and might induce clotting in whole blood and PRP, which makes it rather unsuitable for flow cytometry that is designed for analysis of cells in suspension. Other agonists, such as TRAP-6, would have been more appropriate to be used in both PRP and whole blood as it does not induce clotting.

Investigating activation of platelet GPIIb/IIIa or fibrinogen binding in addition to CD62P would have been suitable as the authors show that the DNA hydrogel induces platelet aggregation.

2) Rather than performing flow cytometry, it would have been interesting to stain surface activation markers of platelets bound to the DNA hydrogel using immunocytochemistry and antibodies against P-selectin and/or activated GPIIb/IIIa (in addition to showing platelet aggregation and pseudopod formation in Figure 4B). Did the authors check for other cells bound to the DNA hydrogel, such as neutrophils, or blood clot components, such as fibrin?

3) RNA Seq data seems overinterpreted. Genes listed under platelet activation genes are not typical candidates for determining platelet activation and could be derived from any cell type in the blood as RNA Seq was performed with whole blood samples. For investigating intracellular signaling in platelets, methods like Western blot using platelet lysates are more appropriate.

4) Figure 6A: the illustration of how the DNA hydrogel band might initiate clotting is very speculative and not substantiated enough by the presented data: e.g. it is not shown that the DNA hydrogel interacts with Factor VII and that this leads to platelet activation; activation of platelet GPIIb/IIIa and fibrinogen binding not examined in this study

5) This study neglected to investigate whether the DNA hydrogel initiates coagulation by activating plasmatic coagulation which then could lead to platelet activation by generated thrombin which is a potent platelet activator. E.g., factors XII and XI (intrinsic pathway) are known to be activated by DNA, contributing to thrombin generation (Matafonov et al Blood 2014), platelets could also be directly activated by DNA (Fiedel et al J Immunol 1979)

6) Could the authors discuss putative side effects when blood and tissue are exposed to the DNA hydrogel? For example, platelet activation results in PF4 release. PF4 is highly positively charged and is known to bind to negatively charged polyanions such as DNA, which could induce immunogenicity and formation of platelet activating antibodies, leading to a risk of thrombotic complications (Jaax et al Blood 2013).

Minor comments:

1) For all data presented as mean \pm SD, please indicate the number of experiment (n=?) in the figure legends. Which sample size was used for the RNA-Seq analysis?

2) Figure 5 shows significant differences in blood loss between groups in all three bleeding models. However, in the Results section, significance is only indicated for the femoral artery injury model (page 10). Please also add p-values for the other two models (tail amputation and liver puncture) in the text.

3) Please provide detailed information in the Methods section on how platelets and erythrocytes were stained in CLSM (Figure 4B).

4) Please explain what "DNA aq" means and why it was used as a control in the flow-cytometric experiments shown in Figure 6. How do the authors explain the difference in P-selectin expression induced by DNAgel or CaCl₂ in whole blood and PRP (much higher in whole blood)?

5) Figure 7C and D: Were the differences in wound healing significant (page 14, first three lines)?

6) Figure 2G: Why was deionized water and not blood used for evaluating the swelling behavior of the DNA hydrogel

Reviewer #3:

Remarks to the Author:

In this manuscript, the authors developed a novel wound hemostatic adjuvant based on DNA hydrogel. The idea of multifunctional NETs-like DNA hydrogel design for wound hemostasis and healing is ingenious and interesting. They found that the DNA hydrogel exhibited super liquid absorbency, stable tissue adhesion and the ability to absorb plasma and activate platelets, offering a comprehensive "physically" and "physiological" hemostatic method. In general, it is a meaningful and interesting study. I would recommend accepting this manuscript in Nature Communications after addressing the following issues.

1. PEGDA and strongly alkaline NaOH were used in the chemical synthesis of DNA hydrogel. please provide the evidence of biocompatibility.
2. The authors need to provide the liquid absorption characteristics of DNA hydrogel in blood or under physiological conditions.
3. Bio-based and bioinspired hydrogels have been widely reported for hemostasis and wound healing. What is the difference and advantage of the NETs-like DNA hydrogel? The comparison and discussion should be extended in this direction.
4. The author should indicate the number of experimental subjects in the caption of figures or experimental methods.
5. The font in the caption of Figure 2 is not consistent, e.g., μm . Some other typo errors should also be carefully checked.
6. The author should explain the constitutes or formulates of the DNA aq that used in Section 2.6.
7. Some pictures or the text in the figures are unclear, e.g., Figure 6C.

DETAILED RESPONSE TO REVIEWS

Dear editors of *Nature Communications*

According to the comments on our manuscript (title “**DNA hydrogel based Wound Hemostatic Adjuvant**”, **Manuscript ID: NCOMMS-23-51146**) from the reviewers, we have taken a very careful consideration to the criticisms and suggestion. We have given the detailed description of the main changes in the revised manuscript point by point. The changes in the revised manuscript have been identified by highlighting with red color.

The responses to the comments of reviewer #1

In this study, the authors demonstrated rapid hemostasis and effective wound healing of DNA-based hydrogels, inspired by neutrophil extracellular traps (NETs), a mesh-type network of DNA and proteins released from cells. Although the experimental data on materials characterization, in vivo hemostasis, and wound healing capability support the authors' claims, the novelty and originality of the study are highly insufficient. Firstly, the authors utilized previously reported DNA-based hydrogels (J. Am. Chem. Soc. 2020, 142, 22, 10114), indicating a lack of novelty in the materials. Additionally, the hemostatic efficacy and coagulation mechanism of DNA-based materials (relevant to polyphosphate materials) have already been reported in other references (Adv. Funct. Mater. 2015, 25, 1270–1278; Macromol. Biosci. 2017, 17, 1600252; Green Chem., 2021, 23, 629-669; Res Pract Thromb Haemost. 2019; 3: 18–25). The reviewer thinks that the only difference in this study from others was a specific biological analysis regarding the signaling pathway of the clotting cascade (Figure 6), but that is not enough to be published in this journal. The manuscript should be transferred to other sister journals. There are specific comments for further improvement.

Response: We appreciate the constructive suggestions from the reviewer. DNA-based biomaterials have attracted a lot of attention in recent years due to the inherent advantages and have proven successful in tissue regeneration applications, such as neurogenesis and functional recovery (Adv. Mater. 2021, 33, 2102428), bone regeneration (PNAS, 2023,120,17, e2220565120) and have been used in many other cases of three-dimensional cell culture (ACS Appl. Mater. Interfaces 2021, 13, 42, 49705), stem cell fishing (J. Am. Chem. Soc. 2020, 142, 7, 3422) and so on, which demonstrates that DNA material is a research hotspot with huge potential biological and clinical prospects. In this work, we utilized a universal, simple, large-scale and low-cost way to prepare a pure-DNA based hydrogel according to the previously reported method (J. Am. Chem. Soc. 2020, 142, 22, 10114) for biomedical applications instead of the substitute for plastics, which is different to that reported in the literature. Compared to the references the reviewer mentioned, we have added the concept of swelling/expanding of hydrogel for physical compression hemostasis and the specific biological analysis including flow cytometry and RNA sequencing transcriptome analysis, under the premise of tissue sealant for hemostasis. Besides, we utilized the naturally extracted DNA from salmon sperm without precisely sequence design, highlighting the versatility of the DNA biomass material. In addition, apart from the hemostatic properties of DNAgel discussed in this paper, our completed research in other studies has further confirmed the biomedical application potential of these natural nucleic acid hydrogels. Particularly, in full-layer skin appendage

regeneration and tissue regeneration in cases of bone defects, DNAgel has demonstrated positive promoting effects. Over the past few decades, numerous DNA-based hydrogels have been precisely designed and engineered for various applications. Further researches into the application of engineered DNAgels in different biomedical fields will contribute to a deeper understanding of this natural high-polymer material. We hope that DNAgel can provide valuable assistance to human health in the future.

< Major issues >

1. Figure 3: The tissue-adhesive strength of DNAgel (~5 kPa) seems too low to overcome the large transformation during isotropic swelling. The authors should compare its strength to other adhesives, not gelatin sponge. The gelatin sponge cannot be a good control, and the authors should compare the adhesiveness and performance of DNAgel to that of clinically useful hemostatic agents, such as fibrin sealants.

Response: We are appreciative of the valuable comment. Actually, we did not compare our DNA hydrogel with gelatin sponge in the main text because the adhesive properties of gelatin sponge are indeed low. Instead, we compared the strength of the DNA hydrogel with GeIMA, an adhesive commonly used in the literatures. In addition, we compared the mechanical strength of DNAgel with the commercial medical glue Compont®, which is mainly composed of N-Butyl cyanoacrylate, and evaluated the cytotoxicity of these materials. As shown in **Figure S2**, the shear strength and interfacial toughness of DNA hydrogel are 5.29 ± 0.80 kPa and 168.57 ± 6.82 J m⁻², respectively, equivalent to those of GeIMA, but significantly lower than those of the commercial glue. However, the commercial glue-conditioned medium exhibited strong cytotoxicity as shown in **Figure S4**. The cell viability of the commercial glue-conditioned medium was nearly 22% after 24-hour cultivation and decreased to 2.4% after 48-hour cultivation, while that of DNAgel was higher than 90%. The tissue adhesion property of the DNA hydrogel mentioned in this study is just one of its distinctive features as a hemostatic agent. In addition to sealing wounds, the DNA hydrogel also showed the potential to biochemically promote coagulation, which is the broader point we want to make in this article.

Figure S2. (A) Lap shear tests and (B) 90-degree peel tests of different materials with rat dorsal skin (1cm×2cm). Error bar, SDs (n=3)

Figure S4. (A) Cell viability and (B) representative fluorescent images of WRL68 human normal liver cells incubated with DNAgel- or commercial glue-conditioned medium. Scale bar = 200 μ m. Error bar, SDs ($n \geq 3$)

2. Figure 4: The authors said that DNAgel could capture many erythrocytes and platelets in vitro. The experiments should be re-conducted for the sample after in vivo hemostasis tests. In addition, the pore size of the dried DNAgel should be quantitatively analyzed to investigate whether RBCs with a large size or platelets can be trapped in the gels by physical absorption or other chemical interactions (e.g., electrostatic forces).

Response: The suggestions from the reviewer have been immensely helpful. After conducting in vivo hemostasis experiments using DNA hydrogel, we have obtained SEM data of the cross section of DNA hydrogel. It has been observed that a large number of red blood cells tend to accumulated on the surface of the DNA hydrogel rather than being trapped inside (**Figure S6**). These findings provide valuable insights into the hemostatic mechanism of DNAgel in vivo.

Figure S6 SEM image of blood cells accumulated on the surface of the DNA hydrogel

rather than being trapped inside.

3. Is there any significant difference in hemostatic performance between a dried bare DNA film (without PEGDA) and the DNAgel?

Response: Thank you for the reviewer's insights, which help us better convey the core of our research. We additionally prepared a DNA film from 6% DNA solution precursor by simple oven-drying, without crosslinked (called "non-crosslinked DNA"), and compared the *in vitro* hemostatic properties with those of GS and DNAgel. The **Figure S5** showed that the BCI of the non-crosslinked DNA was $75.94 \pm 6.39\%$, significantly higher than that of the DNAgel, which suggested the weaker hemostatic capability of the non-crosslinked DNA. This finding underscores the significance of the specific DNAgel formulation and its unique properties in achieving effective hemostasis.

Figure S5. (A) Absorbance of blood clotting samples at different time points measured at 540 nm with UV-vis spectrophotometer and the representative pictures at 30 mins. Scale bar = 1 cm. (B) BCI and representative pictures. Scale bar = 1 cm. Error bar, SDs (n=3).

4. The authors emphasized the importance of the high liquid absorption capacity and the swelling behavior of DNAgel. Nonetheless, the investigation of hemostasis does not represent these characteristics. The authors should note the hemostatic effect due to blood absorption and swelling to emphasize the properties of the DNAgel. Alternatively, the high liquid absorption capability and linear swelling property of the DNAgel should be simply mentioned as properties of the DNAgel. Similarly, during the hemostasis process using DNAgel, more experiments are needed to show that swelling and expansion of the gel prevent blood outflow.

Response: We appreciate the reviewer's suggestion. We have included supplementary experiments and evaluated the *in vitro* hemostatic properties of the non-crosslinked DNA film mentioned in the previous response, which does not have the swelling capacity. These results shown in **Figure S5** clearly demonstrated that the hemostatic capability of this non-expandable DNA film was significantly weaker than DNA hydrogel.

Figure S5. (A) Absorbance of blood clotting samples at different time points measured at 540 nm with UV-vis spectrophotometer and the representative pictures at 30 mins. Scale bar = 1 cm. (B) BCI and representative pictures. Scale bar = 1 cm. Error bar, SDs (n=3).

5. According to the results of flow cytometry, RNA sequencing, and GSEA analysis (Figure 6), platelet activation and coagulation acceleration effects of DNA gel were demonstrated. However, the molecular biological process of it should be explained more detailed. Stephanie et al. [<https://doi.org/10.1073/pnas.0507195103>] reported DNA failed to abrogate the tissue factor pathway inhibitor (TFPI) function, unlike polyphosphate molecules, and Figure 6B showed whole blood with DNA aq was not significantly different from the untreated group. Please explain in more detail "how the negatively charged surface of DNA gel affects the coagulation factor VII signaling pathway" with some results or references.

Response: We thank the reviewer for this critical comment. As reported by Stephanie et al. [<https://doi.org/10.1073/pnas.0507195103>], our experimental results also did not reveal any clotting effects of DNAaq, which corroborates the dual-coagulation mechanism of DNAgel that we intend to elucidate, involving both "physical" and "biochemical" pathways. Firstly, in the SEM images of **Figure 4A**, it can be observed that DNAgel, even after swelling, exhibits a dense and less porous surface morphology compared to the porous and loose GS. This structural characteristic leads to the entrapment of a significant number of red blood cells and platelets on the material's surface, forming accumulations. Therefore, at the interface between the bleeding wound and DNAgel, the abundant accumulation of dehydrated red blood cells and platelets provides the material basis for the rapid clotting cascade, including factors like fibrinogen and clotting factors, as shown in **Figure 6B-C**. Moreover, when experiments were conducted using non-crosslinked DNA solution, no significant difference in clotting rate was observed compared to the control group, as demonstrated in **Figure 6B-C**. Similarly, when comparing the platelet activation ratio of whole blood and platelets with DNAgel group, a significant increase in platelet activation ratio was observed after co-incubation with DNAgel. Therefore, it is reasonable to conclude that DNAgel's exceptional swelling capacity and dense surface morphology provide a "physical" pathway to accelerate hemostasis.

Furthermore, we aim to elucidate the early molecular events when DNA hydrogel comes into contact with a bleeding wound. To begin, we conducted RNA sequencing (RNAseq) on samples treated with different groups.

6. In the wound healing section, the enlarged position of the histology images is different for each group. Because there are differences in the progress of regeneration between the edges and the central lesion of the damaged area, the authors need to show images from the same location as much as possible.

Response: Thank you for this insightful comment and suggestion of the publications. We modified Figure 7F based on the comments. In tissue section staining, we primarily selected two time points, Day 7 and Day 10, for analysis. At Day 7, none of the groups exhibited complete wound closure; hence, we chose to magnify the initial wound edge location to compare the pathological features of tissue regeneration at that site. From the experimental results, it is evident that, in comparison to the GS group and Control group, the DNAgel group displayed fewer inflammatory cells at the wound edge on Day 7, indicating an earlier transition to the tissue regeneration phase. Notably, there was minimal new epidermal tissue observed in the other two groups. Of significance, in the DNAgel group, we observed neovascularization and regenerated epithelium, collectively suggesting that DNAgel treatment accelerated wound healing in the early stages of wound closure.

After 10 days of treatment, the defective skin in the DNAgel group had completely healed, resembling normal skin in appearance. New hair follicles and sebaceous glands were visible in the histological sections, indicating full regeneration of the epidermal tissue. Therefore, we chose to magnify the mid-wound location to illustrate the differences in direct wound healing capacity among the groups. In the Control group and GS group, there were still numerous inflammatory cells and unclosed wounds.

Furthermore, Masson's trichrome staining and Sirius Red staining demonstrated that the DNAgel group exhibited more deposited regenerative collagen within the wound site compared to the other two treatment groups, further confirming the superior wound healing efficacy of DNAgel.

Figure 7. (F) Hematoxylin and Eosin staining, Masson's Trichrome staining and Sirius Red staining on day 7 and 10 of the newly regenerated skin tissues for each group. The purple dotted line marks the initial trauma margin. Scale bar = 1 mm (top); Scale bar = 100 μm (bottom, enlarged)

< Minor issues >

5. The "Experimental Section" should be checked and explained in more detail. For example, the authors do not provide the rheological measurement procedure (Figure 2D). Also, in other parts, the experimental procedure should be re-written to provide details.

Response: We appreciate for the suggestion, and added detailed experimental methods about physical and chemical characterization of DNAgel in Experimental section, including SEM, rheological measurement, FTIR, histopathological study and el at.

6. Recent references regarding hemostasis of DNA-based materials should be cited (Adv. Funct. Mater. 2015, 25, 1270–1278; Macromol. Biosci. 2017, 17, 1600252).

Response: Thank you for your kindly recommend. We have revised the manuscript and added these related references in the main text.

7. Was the DNA gel used for investigating rheological properties (Figure 2D) measured after fully swelling? If not, it should be suggested to ensure *in vivo* stability.

Response: We appreciate the suggestion made by the reviewer. We evaluated the rheological properties of DNA hydrogel after fully swelling in the PBS and also the heparinized fresh whole blood. As shown in the **Figure S7**, the DNA hydrogel well-swollen in PBS maintained a solid-like behavior, because the storage modulus (G') of it was higher than the loss modulus (G'') at the low strain range (0.1-10%). The gel swollen in PBS getting collapsing as the strain was larger than 35%, and that of the gel swollen in fresh whole blood was larger than 260%, both of which were higher than the highest strain in the body (10%), ensuring the stability of DNAgel *in vivo*.

Figure S7. G' and G'' of DNAgel that well-swollen in (A) PBS and (B) fresh whole blood on strain amplitude sweep.

8. Can rheological properties and swelling behaviors be changed due to swelling or adhesion of blood components (e.g., RBCs, platelets, plasma) instead of DW?

Response: We investigated the swelling behaviors in PBS and fresh whole blood and assessed the rheological properties of well-swollen DNAgel. First of all, the DNA hydrogel could swell approximately 14.3 times its own weight in PBS within 4 hours, and the swelling ratio of DNAgel in blood could also reached to 15.5-fold (**Figure S1**). Secondly, DNA hydrogel swollen in PBS or maintained a solid-like behavior at the low strain range (0.1-10%), while DNAgel swollen in blood exhibited a higher modulus, improving the rigidity and ductility of DNAgel (**Figure S7-8**). Nevertheless, the DNAgel could still maintain mechanical stability *in vivo* because the collapsing points were higher than the highest strain in the body (10%).

Figure S1. Swelling ratio of DNAgel in (A) DW, (B) PBS, and (C) fresh whole blood.

Figure S7. G' and G'' of DNAgel that well-swollen in (A) PBS and (B) fresh whole blood on strain amplitude sweep.

Figure S8. G' and G'' of DNAgel that well-swollen in (A) PBS and (B) fresh whole blood on frequency sweep.

9. The authors mentioned that the commercially available absorbable gelatin sponge (GS) is not good at blood absorption in the Discussion section. To demonstrate it, its contact angle (Figure S6) at various time scales should be compared with that of DNA gel (Figure 2F).

Response: Thank you for the kind suggestion. We sorted out the contact angle measurement data of DNAgel and GS, and presented them as follows. The water drop could be quickly absorbed by DNAgel once contacted to the surface of DNAgel, and the contact angle rapidly reduced to 15.3° within 90 seconds. In contrast, the contact angle was still greater than 90° after the water dropped onto the GS for 90 seconds, as shown in **Figure S16**.

Figure 2F. Images of water droplet absorbed by DNAgel. Scale bar = 1 mm

Figure S16. Images of contact angle measurement of GS. Scale bar = 1 mm

10. Please check entire manuscript for typos.

For example,

Page 17, Lines 266-271 "~pathways: i . SFKs activated SYK~" should be "~pathways:

i . SFKs activated SYK~".

Figure 7E, color labels were missing

...

Response: Thanks for your detailed reminders. We've checked entire manuscript and corrected the typos, such as "Intracellular signaling may be transmitted by two pathways: i. SFKs activated SYK and promoted the assembly of the SLP76/LAT/Btk/Vav complex, which mediated PLC γ 2 activation in a manner similar to the GPVI-mediated ITAM signaling pathway; ii. SFKs ~ ". We also added the color label of immunohistochemistry staining image in **Figure 7E**, shown as below.

Figure 7. (E) Representative images of immunohistochemistry staining with CK14 on day 7 (Wound closure time points in DNAgel group) and day 10 (Total skin regeneration time point in DNAgel group) for each group. Scale bar = 100 μ m.

The responses to the comments of reviewer #2

The authors aimed at developing a novel, DNAgel band-aid with strong hemostatic properties. The DNA hydrogel band-aid was adhesive, procoagulant, biocompatible, and promoted wound healing. The biophysical properties of the DNA hydrogel band-aid were examined, showing in three rat bleeding models that it prevented blood loss. The DNA hydrogel band-aid was also biocompatible in in vitro cell culturing and accelerated wound healing in an in vivo skin incision model.

As a major drawback, the procoagulant activity was mainly mediated platelet activation - which was based on flow cytometric and RNA-Seq analysis neglecting plasmatic coagulation.

Major comments:

1) The flow cytometry study was not well performed, and data analysis is questionable: Figure 6B: Untreated samples of PRP and whole blood show already 50% preactivated platelets (acceptable would be lower than about 10%)! In this regard, PRP generation should be performed at RT and not at 4 °C (stated in the Methods, page 18) as platelets are sensitive to cold and could get preactivated.

Figure 6C: x-axis and y-axis labeling of all plots (CD62P/CD61) and quadrant labeling below untreated is not correct (CD62P+ is determined as inactive platelets! CD62P is a platelet activation marker); according to the Methods, the x-axis should be CD61 (FITC Mouse Anti-Rat CD61), and y-axis should be CD62P. How was the gating strategy?

All platelets should have been CD61-FITC-positive as this is a platelet lineage marker (expressed on the surface without and with activation), but the representative plots provided in Figure 6C only show marginal FITC-positivity. Were the antibodies titrated for saturating concentrations? Were isotype control antibodies used to distinguish between specific and unspecific antibody binding?

The use of CaCl₂ might not be an ideal positive control for analyzing P-selectin expression as it activates the coagulation cascade and might induce clotting in whole blood and PRP, which makes it rather unsuitable for flow cytometry that is designed for analysis of cells in suspension. Other agonists, such as TRAP-6, would have been more appropriate to be used in both PRP and whole blood as it does not induce clotting.

Investigating activation of platelet GPIIb/IIIa or fibrinogen binding in addition to CD62P would have been suitable as the authors show that the DNA hydrogel induces platelet aggregation.

Response: Your suggestions are important for further exploration of our materials. We extracted fresh blood from rats based on your comments and prepared platelet-rich plasma at room temperature within 10 minutes of extraction and performed flow cytometry on activated platelets in DNAgel-treated plasma and/or whole blood. The active platelets (CD61⁺/CD62P⁺ cells) were quantified again. The agonist of the positive control group was changed to TRAP-6, while GS, DNA aq-treated and untreated platelet-rich plasma and/or whole blood were used as controls. In the staining step of flow cytometry, we used the CD61 antibody reached the saturation of 2.0 µg per 10⁶ cells in 100 µl volume, while the

suggested use of this reagent is 1.0 μg per 10^6 cells in 100 μl volume. The x-axis represents FITC, which is the intensity of CD61, and the y-axis represents PE, which is the intensity of CD62P. It was shown in **Figure 6B-C** that 9.78% of particles in the platelet rich plasma were showed FITC positivity, and 70.75% platelets were activated in the DNAgel group, second only to the positive control (74.06%), and higher than the group of GS (63.74%), DNA aq (53.61%) and untreated groups (9.20%). In addition, 64.95% of the platelets in the whole blood were activated, comparable to the positive control group (69.20%) and significantly higher than the other three groups. At the same time, we also provide a gating strategy for determining FITC and PE-positive cells during flow cytometry, as shown in the **Figure S9**. Furthermore, we lysed the platelets adhered to the DNAgel and measured the GPIIb/IIIa protein content in the lysate using Elisa kit. It was shown that the content of GPIIb/IIIa complex of DNAgel-treated group was significantly higher than that of the untreated group (**Figure S10**), indicating the activation and aggregation of platelets on the surface of DNA hydrogel.

Figure 6. DNAgel promoted platelet activation and accelerated the clotting process by upregulating related genes. (A) Signaling pathway of clotting cascade triggered by DNAgel.

(B) Statistical results of platelet activation and (C) Representative flow cytometry plots under different treatment conditions. Error bar, SDs ($n \geq 3$). (D) RNA-seq analysis of whole blood cells, and differential expressed genes presented in a volcano map. Cutoff line: Log_2 fold change (Log_2FC) $> |2|$ and adjusted p value < 0.01 . (E) Heatmap of differential expressed genes related to platelet activation and (F) GSEA of KEGG pathway analysis related to platelet.

Figure S9. Exemplary flow cytometry gating. (A) gating strategy for unstained cells (B) representative gating for FITC Mouse Anti-Rat CD61 and PE Mouse Anti-Rat CD62P stained cells in TRAP-6 treated group.

FigureS10. GPIIb/IIIa protein content of platelets adhered to the DNAgel.

2) Rather than performing flow cytometry, it would have been interesting to stain surface activation markers of platelets bound to the DNA hydrogel using immunocytochemistry and antibodies against P-selectin and/or activated GPIIb/IIIa (in addition to showing platelet

aggregation and pseudopod formation in Figure 4B). Did the authors check for other cells bound to the DNA hydrogel, such as neutrophils, or blood clot components, such as fibrin?

Response: Thank you for your suggestion. We used the Elisa kit of GPIIb/IIIa to explore the expression of platelet surface marker GPIIb/IIIa after DNAgel treatment. We lysed the platelets adhered to the material and measured the GPIIb/IIIa protein content in the lysate using Elisa kit. The results showed that after DNAgel treatment, the content of GPIIb/IIIa complex on the platelet surface was significantly higher than that in the untreated group, proving that our DNAgel can successfully activate the GPIIb/IIIa complex on the platelet surface, thereby inducing platelet aggregation.

Figure S10. GPIIb/IIIa protein content in the platelets adhered to the DNAgel.

3) RNA Seq data seems overinterpreted. Genes listed under platelet activation genes are not typical candidates for determining platelet activation and could be derived from any cell type in the blood as RNA Seq was performed with whole blood samples. For investigating intracellular signaling in platelets, methods like Western blot using platelet lysates are more appropriate.

Response: Thank you for your suggestion. Based on the RNAseq results, we extracted the RNA and protein of platelets after DNAgel treatment and verified them by quantitative polymerase chain reaction (qPCR) and western blot. The results showed that after DNAgel treatment, the RNA expression of VAV2, VAV3 and PLCG2 in platelets increased significantly (**Figure S11**). At the same time, the results of western blot also confirmed this conclusion. These results can further support our speculation on the mechanism of DNAgel promoting coagulation, which is the SFKs activated SYK and promoted the assembly of the SLP76/LAT/Btk/Vav complex, which mediated PLC γ 2 activation in a manner similar to the GPVI-mediated ITAM signaling pathway.

Figure S11. (A) PLCG2, Vav2 and Vav3 mRNA detected in platelets treated with DNAgel. (B) Western blot results for the identification of PLCG2, Vav2 and Vav3 protein expression in platelets treated with DNAgel. GAPDH is used as reference protein.

4) Figure 6A: the illustration of how the DNA hydrogel band might initiate clotting is very speculative and not substantiated enough by the presented data: e.g. it is not shown that the DNA hydrogel interacts with Factor VII and that this leads to platelet activation; activation of platelet GPIIb/IIIa and fibrinogen binding not examined in this study

Response: Thank you for your valuable opinions. Your opinions are important for further exploring the coagulation mechanism of our materials. We have described the use of Elisa kit to explore the expression of platelet surface marker GPIIb/IIIa after DNAgel treatment (**Figure S10**). Furthermore, we used the Factor VII Elisa kit to stain and quantified the OD value of coagulation factor VII (F7) in DNAgel-treated plasma, while untreated plasma was used as control. The results showed that the F7 content in the plasma of the DNAgel-treated group decreased significantly compared with the untreated group (**Figure S12**). We believe that this is due to the activation of FSAP by negatively charged DNAgel. FSAP has been reported to be attracted and activated by negatively charged surfaces or molecules (Muhl, L., et al. The FEBS Journal, 2009), then activating and binding to F7 (Römisch, J., et al. Blood coagulation & fibrinolysis, 1999). Therefore, FSAP activated by DNAgel bound more F7 to the surface of the material. After the material was taken out of the treated plasma, there was a significant decrease in F7 in the plasma.

Figure S10. GPIIb/IIIa protein content in the platelets adhered to the DNAgel.

Figure S12. F7 content in the DNAgel-treated plasma.

5) This study neglected to investigate whether the DNA hydrogel initiates coagulation by activating plasmatic coagulation which then could lead to platelet activation by generated thrombin which is a potent platelet activator. E.g., factors XII and XI (intrinsic pathway) are known to be activated by DNA, contributing to thrombin generation (Matafonov et al Blood 2014), platelets could also be directly activated by DNA (Fiedel et al J Immunol 1979)

Response: Your suggestion provided us with a new idea, so we used the thrombin Elisa kit to detect the thrombin content in plasma treated with DNAgel. It can be seen from the results that after DNAgel treatment for 15 minutes, the thrombin content in the plasma increased significantly compared with the untreated group (**Figure S13**), which shows that DNAgel can also initiate coagulation through the plasma coagulation pathway.

Figure S13. Thrombin content in the DNAgel-treated plasma.

6) Could the authors discuss putative side effects when blood and tissue are exposed to the DNA hydrogel? For example, platelet activation results in PF4 release. PF4 is highly positively charged and is known to bind to negatively charged polyanions such as DNA, which could induce immunogenicity and formation of platelet activating antibodies, leading to a risk of thrombotic complications (Jaax et al Blood 2013).

Response: Thank you for your suggestion, it will be of great help for us to further study the biosafety of the materials we manufacture. In recent years, due to the global pandemic of COVID-19, thrombosis in the human body caused by PF4 has indeed become a focus of concern. But for our material, it does not need to be injected into the human blood circulation like traditional procoagulant factor treatment. On the contrary, it activates

coagulation factors, absorbs blood, and further adheres platelets and red blood cells to the surface of the material, which is used to perform in-situ hemostasis at the wound site. This hemostatic effect does not spread throughout the body. In addition, the rheological properties of the material show that the hemostatic hydrogel we made can still maintain a stable solid form in the body, which can also prevent the material from spreading throughout the body due to rheological defects due to blood circulation. In conclusion, we believe that even if the material causes PF4 to be released from activated platelets, the amount is minimal and does not diffuse throughout the human circulatory system.

Minor comments:

7) For all data presented as mean \pm SD, please indicate the number of experiment (n=?) in the figure legends. Which sample size was used for the RNA-Seq analysis?

Response: Thanks for your kind suggestions. We've indicated the number of experiments in the figure legends. As for the sample size of the RNA-seq analysis, n=3.

8) Figure 5 shows significant differences in blood loss between groups in all three bleeding models. However, in the Results section, significance is only indicated for the femoral artery injury model (page 10). Please also add p-values for the other two models (tail amputation and liver puncture) in the text.

Response: Thanks for your suggestion. We've added t p-values for the tail amputation and liver puncture models in the text.

9) Please provide detailed information in the Methods section on how platelets and erythrocytes were stained in CLSM (Figure 4B).

Response: We've added the detailed methods on how to stain platelets and erythrocytes in CLSM (Figure 4B) in the experimental section. For the immunofluorescence, the DNAgel was immersed in fresh anticoagulated whole blood until the DNAgel was fully swollen. It was then washed in PBS buffer containing 1% BSA to remove blood cells that did not adhere to the DNAgel. The DNAgel was then fixed in 4% paraformaldehyde. Mouse Anti-Human CD61 (BD Biosciences) was used as primary antibody to stain platelets, and accordingly FITC Conjugated Rabbit Anti-Mouse IgG Rabbit Polyclonal Antibody (Huabio) was used as secondary antibody. After platelet staining, Hoechst 33342 (Thermo fisher) was used to stain the DNA. Due to the spontaneous red fluorescence of its hemoglobin, red blood cells can be directly observed under RFP excitation light without staining.

10) Please explain what "DNA aq" means and why it was used as a control in the flow-cytometric experiments shown in Figure 6. How do the authors explain the difference in P-selectin expression induced by DNAgel or CaCl₂ in whole blood and PRP (much higher in whole blood)?

Response: Thank you for your suggestion. DNA aq represents non-crosslinked DNA solution. The purpose of using this component as one of the controls is to prove that in addition to the negative charge of DNA, the three-dimensional structure of the DNAgel we prepared is useful for capturing platelets, red blood cells and enrichment

of coagulation factors in blood are more important for hemostasis. As for the difference in P-selectin expression induced by DNAgel or CaCl₂ in whole blood and PRP, it may be because we used samples from different animals when performing flow cytometry on whole blood and PRP, and these blood samples having been stored at 4°C for a period, there is varying degrees of platelet pre-activation. We have updated the flow cytometry results in Reviewer #2's Q1. The samples for this result were from the same individual, and the experiment started within 10 minutes after the blood was drawn. The entire experiment was conducted at room temperature.

11) Figure 7C and D: Were the differences in wound healing significant (page 14, first three lines)?

Response: Thanks for your suggestion, we have added the significance analysis results to the figure.

Figure 7. (C) Wound size and (D) Healing rate at different days for each group, according to $Extent\ of\ healing\ (\%) = \left(1 - \frac{A_t}{A_0}\right) \times 100\%$. Error bar, SDs ($n \geq 3$). (* $p < 0.05$, ** $p < 0.01$, *** $p < 0.001$, and **** $p < 0.0001$).

12) Figure 2G: Why was deionized water and not blood used for evaluating the swelling behavior of the DNA hydrogel

Response: Thank you for your question. We used deionized water to evaluate the maximum swelling capacity of DNAgel. We also supplemented the swelling data of DNAgel in PBS and whole blood in **Figure S1**. The DNA hydrogel could swell approximately 14.3 times its own weight in PBS within 4 hours. Similarly, the DNA hydrogel was also able to absorb liquid up to 15.5 times its own weight in blood, although the clot formed and adhered to the surface of the DNA hydrogel as the fluid was absorbed.

Figure S1. Swelling ratio of DNAgel in (A) DW, (B) PBS, and (C) fresh whole blood.

The responses to the comments of reviewer #3

In this manuscript, the authors developed a novel wound hemostatic adjuvant based on DNA hydrogel. The idea of multifunctional NETs-like DNA hydrogel design for wound hemostasis and healing is ingenious and interesting. They found that the DNA hydrogel exhibited super liquid absorbency, stable tissue adhesion and the ability to absorb plasma and activate platelets, offering a comprehensive “physically” and “physiological” hemostatic method. In general, it is a meaningful and interesting study. I would recommend accepting this manuscript in Nature Communications after addressing the following issues.

1. PEGDA and strongly alkaline NaOH were used in the chemical synthesis of DNA hydrogel. please provide the evidence of biocompatibility.

Response: Thank you for this kind suggestion. Though PEGDA and NaOH were used for DNAgel crosslinking, the untreated chemicals were also removed by washing in the deionized water and phosphate-buffered saline before using. We've incubated WRL68 human normal liver cells with DNAgel-conditioned medium and provided the representative fluorescent images and the results of *in vitro* cell viability, which showed the good biocompatibility of our DNAgel (**Figure S4**). The H&E staining of the major organs of DNAgel group also showed no significant difference from that of control group, indicating the excellent biocompatibility of DNAgel (**Figure S15**).

Figure S4. (A) *In vitro* cell viability and (B) representative fluorescent images of WRL68 human normal liver cells incubated with DNAgel- or commercial glue-conditioned medium. Scale bar = 200 μm. Error bar, SDs (n≥3).

Figure S15. Images of major organ tissue slices with H&E staining in Normal and DNAgel groups.

2. The authors need to provide the liquid absorption characteristics of DNA hydrogel in blood or under physiological conditions.

Response: We appreciate the suggestion made by the reviewer. We evaluated the swelling ratio of DNA hydrogel in PBS and also the heparinized fresh whole blood. As shown in the **Figure S1**, the DNA hydrogel could swell approximately 14.3 times its own weight in PBS within 4 hours. Similarly, the DNA hydrogel was also able to absorb liquid up to 15.5 times its own weight in blood, although the clot formed and adhered to the surface of the DNA hydrogel as the fluid was absorbed.

Figure S1. Swelling ratio of DNAgel in (A) DW, (B) PBS, and (C) fresh whole blood.

3. Bio-based and bioinspired hydrogels have been widely reported for hemostasis and wound healing. What is the difference and advantage of the NETs-like DNA hydrogel? The comparison and discussion should be extended in this direction.

Response: We are appreciative of your suggestion and revised the related discussion in the main text. Neutrophil extracellular traps (NETs) is a meshwork of extracellular DNA released and precipitated fibers combined with histones and other proteins, which was proven to promoted coagulation at the bleeding sites. In this paper, we prepared the NETs-like DNA hydrogel from naturally extracted salmon sperm DNA using a simple chemical cross-linking method, demonstrating the advantages of wide sources and easy accessibility. Besides, NET-like DNA could promote hemostasis and wound repair with good biocompatibility, similar to other bio-based and bioinspired hydrogels as previous reported. What is more important, owe to this kind of three-dimensional structure similar to the natural NET, the DNAgel promoted the occurrence of coagulation at the molecular level.

4. The author should indicate the number of experimental subjects in the caption of figures or experimental methods.

Response: Thanks for the kind suggestion. We've added the number of experiments in the figure legends.

5. The font in the caption of Figure 2 is not consistent, e.g., μm . Some other typo errors should also be carefully checked.

Response: We appreciate the reviewer's suggestion, and have checked entire manuscript and corrected the typo errors.

6. The author should explain the constitutes or formulates of the DNA aq that used in Section 2.6.

Response: DNA aq represents non-crosslinked DNA solution. We set the DNA aq as a control group in order to assess the effect of the fluid absorbed by DNAgel on promoting coagulation cascade reaction. Additionally, the three-dimensional structure of the DNAgel we prepared is useful for capturing platelets, red blood cells and enrichment of coagulation factors in blood are more important for hemostasis.

7. Some pictures or the text in the figures are unclear, e.g., Figure 6C.

Response: Thanks for the comment and we've corrected the Figure 6 and improved the clarity of the figure.

Reviewers' Comments:

Reviewer #1:

Remarks to the Author:

In this manuscript, the authors have developed DNA hydrogels as hemostatic agents and have conducted a detailed analysis of the mechanisms underlying their effective sealing properties. Despite efforts to address reviewer concerns, the arguments presented remain unpersuasive for the following reasons:

1. Regarding novelty, the authors initially stated that "DNAgel has demonstrated positive promoting effects, particularly in full-layer skin appendage regeneration and tissue regeneration in cases of bone defects." However, they have not provided specific results regarding bone tissue regeneration following treatment with DNAgel as a hemostatic agent. While the authors highlight the naturally derived nature of DNAgel from salmon sperm DNA, it's noted that such naturally derived DNAgel had previously been fabricated, as referenced in "Adv. Funct. Mater. 2015, 25, 1270–1278 and many papers mentioned in Adv. Funct. Mater. 2024,34, 2309070."

2. One major issue concerns the comparison of adhesive strength between DNAgel and cyanoacrylate-based glue, with the authors asserting that DNAgel offers better cytocompatibility compared to commercial glue. While it's acknowledged that naturally derived hydrogels may exhibit greater biocompatibility than cyanoacrylate, it's essential to consider that cyanoacrylate-based glue has not yet been approved for clinical use onto internal organs due to cytotoxicity concerns. Therefore, the authors should compare DNAgel with other biocompatible sealants (e.g., Tisseal, Coseal, or fibrin glues).

3. Furthermore, the reviewer raises concerns regarding the weak adhesiveness of DNAgel and its potential limitations in achieving wound closure over large areas. It's crucial to establish the maximum area for wound closure using DNAgel. The current in vivo setting only includes tests using wounds with very small apertures (~6 mm). It can be also demonstrated in large animal models (e.g., pig or rabbit). (Nature Communications volume 10, Article number: 2060 (2019))

Reviewer #2:

Remarks to the Author:

1) The authors followed the suggestion to change the flow cytometric experiments. They improved the PRP preparation method and used a suitable agonist (TRAP-6 instead of CaCl₂) as a positive control for proper platelet activation, resulting in expected values of P-selectin expression in unstimulated and stimulated samples. They also provided the gating strategy. However, it is still unclear why not all platelets (100%) stain positive for the platelet lineage marker CD61 (10-30% only), and the gating strategy is questionable. Platelets were gated based on FSC/SSC characteristics (followed by single cell gating) but should have been gated based on a platelet lineage marker such as CD61 (if staining works properly), especially in whole blood samples. It needs to be mentioned what is shown in Figure S9, PRP or whole blood?

The authors misunderstood the suggestion to analyze activation of GPIIb/IIIa. It was meant to measure the conformational change of GPIIb/IIIa from the low-affinity into the high-affinity conformation to allow for efficient fibrinogen binding. This can be measured by flow cytometry using a specific antibody clone that only binds to the activated form of GPIIb/IIIa on the surface of platelets. However, the authors checked for the protein content of total GPIIb/IIIa after platelet lysis and by ELISA, which does not allow to distinguish between the low-affinity and high-affinity state.

2) Instead of using proposed methods such as immunohistochemistry and suitable antibodies, the authors respond to this comment by referring again to the measurement of total GPIIb/IIIa by

ELISA, which is not appropriate to answer the question raised. In addition, the authors did not comment on whether other cell types or blood clot components were bound to the DNA hydrogel.

3) The authors confirmed the upregulation of some candidates found in RNA Seq data of whole blood samples with qPCR and Western blot using RNA and protein derived from isolated platelets as suggested.

4) Referring a third time to total GPIIb/IIIa content and the same data as before is not satisfying. However, the authors now present new data indirectly showing that FVII might bind to the DNA hydrogel but neglecting the activation status of FVII.

5) The authors can now demonstrate that the hydrogel might initiate plasmatic coagulation by presenting increased levels of thrombin in DNA hydrogel-treated plasma.

6) The authors' response is plausible.

7) Done for the figures of the initial version of the manuscript but not provided in newly added Figures S1 and S10-S14.

8) Done.

9) Done.

10) Done.

11) Done.

12) Done.

Reviewer #3:

Remarks to the Author:

After revision, I think the manuscript can now be accepted for publication

DETAILED RESPONSE TO REVIEWS

Dear editors of *Nature Communications*

According to the comments on our manuscript (title “**DNA hydrogel based Wound Hemostatic Adjuvant**”, **Manuscript ID: NCOMMS-23-51146**) from the reviewers, we have taken a serious consideration to the criticisms and suggestion. We accept the opinions from the reviewers and have given the detailed description of the main changes in the revised manuscript point by point. The changes in the revised manuscript have been identified by highlighting with red color.

The responses to the comments of reviewer #1

In this manuscript, the authors have developed DNA hydrogels as hemostatic agents and have conducted a detailed analysis of the mechanisms underlying their effective sealing properties. Despite efforts to address reviewer concerns, the arguments presented remain unpersuasive for the following reasons:

1. Regarding novelty, the authors initially stated that "DNAgel has demonstrated positive promoting effects, particularly in full-layer skin appendage regeneration and tissue regeneration in cases of bone defects." However, they have not provided specific results regarding bone tissue regeneration following treatment with DNAgel as a hemostatic agent. While the authors highlight the naturally derived nature of DNAgel from salmon sperm DNA, it's noted that such naturally derived DNAgel had previously been fabricated, as referenced in "Adv. Funct. Mater. 2015, 25, 1270–1278 and many papers mentioned in Adv. Funct. Mater. 2024,34, 2309070."

Response: Thank you for your insights and for highlighting the aspects of novelty in our manuscript titled "DNA Hydrogel Based Wound Hemostatic Adjuvant." We appreciate your detailed examination and understand the importance of clearly presenting the innovative elements of our research.

Regarding the point on bone tissue regeneration, we are very sorry as we may have been inaccurate in expressing what we meant. We meant to illustrate the potential future application of DNAgel in downstream medical issues for example bone tissue regeneration.

Many thanks for bringing up the references which we have cited. We know of the well cited "Adv. Funct. Mater. 2015, 25, 1270–1278" work from Prof Haeshin Lee group and in fact we in the biomaterials field have learnt much from one of the earliest papers on DNA hydrogels. But there are key differences between the Lee's DNA hydrogel and our DNAgel. Based on our interpretation of the AFM pp, especially in Figure 5 and Figure 7 is that the Lee group really proved that the Tannic Acid was the all important "molecular glue" in their DNA hydrogel and explained the molecular mechanism of tannic acid (Figure 2, AFM 2015; Figure 5, AFM 2015) in creating the molecular level integrity of the DNA hydrogel. In our case, we are pre-preparing the DNA strands and then inducing fast gelation with PEGDA as a cross-linker. Even though, in both cases, DNA hydrogel was used but under the hood, molecularly speaking, both technologies are different. You are absolutely right that we are not the only other paper describing the use of DNA hydrogels for other important biomedical

applications but as far as hemostasis and wound healing, DNAgel is the first hydrogel that combined first ultrafast and effective hemostasis followed up by enhanced wound healing. Moreover, we also worked out the cell biology mechanism level of the hemostatic process which may open up future avenue in the future to further tweak at least the hemostatic process through working with the clotting process machinery.

We value your suggestions and are committed to enhancing our manuscript to meet the high standards of Nature Communications. We look forward to further constructive feedback and are eager to refine our work to contribute meaningfully to the field.

2. One major issue concerns the comparison of adhesive strength between DNAgel and cyanoacrylate-based glue, with the authors asserting that DNAgel offers better cytocompatibility compared to commercial glue. While it's acknowledged that naturally derived hydrogels may exhibit greater biocompatibility than cyanoacrylate, it's essential to consider that cyanoacrylate-based glue has not yet been approved for clinical use onto internal organs due to cytotoxicity concerns. Therefore, the authors should compare DNAgel with other biocompatible sealants (e.g., Tisseal, Coseal, or fibrin glues).

Response: We appreciate the reviewer's concern about the adhesive mechanical properties of our DNAgel for blood clotting and wound healing. We have removed the reference to cyanoacrylate based glue and pivoted over to biocompatible sealants. As you suggested, we compared the adhesive strength between our DNAgel and the Fibrin-based hemostatic agent (FM) (Int. J. Mol. Sci. 2024, 25(7), 4069). It was found that the adhesive strength of FM was 2.57 ± 0.47 kPa, which is on the same order of magnitude as that of DNAgel (5.29 ± 0.80 kPa), and even less than the latter. It is undeniable that adhesive strength is a crucial attribute of sealants and our manuscript aims to propose a bionic hemostatic agent with the abilities of tissue adhesion, swelling, enrichment of red blood cells and activation of platelets, which integrally contributes to blood clot formation.

3. Furthermore, the reviewer raises concerns regarding the weak adhesiveness of DNAgel and its potential limitations in achieving wound closure over large areas. It's crucial to establish the maximum area for wound closure using DNAgel. The current in vivo setting only includes tests using wounds with very small apertures (~6 mm). It can be also demonstrated in large animal models (e.g., pig or rabbit). (Nature Communications volume 10, Article number: 2060 (2019))

Response: Thank you for your insightful comments regarding the adhesiveness of DNAgel and its application over larger wound areas. We agree with your very good point to test the material in larger animal models, which indeed could enrich the translational aspect of our research.

Our manuscript primarily aims to introduce the fundamental properties of DNAgel and its potential in hemostatic applications, particularly focusing on its performance in skin wound closure and deep, non-compressible hemorrhagic sites. The versatility of DNAgel allows for customization in size and shape to match specific wound dimensions, which is a

significant advantage for clinical applications. In the current study, we designed experiments on rats to demonstrate DNAgel's efficacy in critical-sized skin defects and classic hemorrhage models, including tail amputation and femoral artery bleeding. These models were chosen to validate the rapid hemostatic properties of DNAgel on bleeding wound surfaces and its capacity to facilitate coagulation, under conditions of acute hemorrhage.

So we fully agree with the value of extending our studies to include larger animal models such as pigs or rabbits, as this could further substantiate the scalability and effectiveness of DNAgel across a broader range of clinical scenarios. Indeed, incorporating such models is part of our future research plans. We believe that our current experimental results sufficiently support our hypotheses and conclusions, demonstrating the promise of DNAgel in hemostatic applications.

We are grateful for your recommendations and look forward to exploring these additional dimensions in our ongoing research. Your suggestions are instrumental in guiding the future direction of our work toward clinical translation.

The responses to the comments of reviewer #2

1) The authors followed the suggestion to change the flow cytometric experiments. They improved the PRP preparation method and used a suitable agonist (TRAP-6 instead of CaCl₂) as a positive control for proper platelet activation, resulting in expected values of P-selectin expression in unstimulated and stimulated samples. They also provided the gating strategy. However, it is still unclear why not all platelets (100%) stain positive for the platelet lineage marker CD61 (10-30% only), and the gating strategy is questionable. Platelets were gated based on FSC/SSC characteristics (followed by single cell gating) but should have been gated based on a platelet lineage marker such as CD61 (if staining works properly), especially in whole blood samples. It needs to be mentioned what is shown in Figure S9, PRP or whole blood?

The authors misunderstood the suggestion to analyze activation of GPIIb/IIIa. It was meant to measure the conformational change of GPIIb/IIIa from the low-affinity into the high-affinity conformation to allow for efficient fibrinogen binding. This can be measured by flow cytometry using a specific antibody clone that only binds to the activated form of GPIIb/IIIa on the surface of platelets. However, the authors checked for the protein content of total GPIIb/IIIa after platelet lysis and by ELISA, which does not allow to distinguish between the low-affinity and high-affinity state.

Response: We appreciate the constructive suggestions from the reviewer and adjust the gating strategy of flow cytometry. Here, taking whole blood samples as examples, platelets were gated based on FSC/SSC characteristics, single cell gating, and platelet lineage marker CD61 gating in sequence, followed by the CD62P gating to distinguish the active/inactive stage, as shown in **Figure S11**. The statistical results of platelet activation and representative flow cytometry plots under different treatment conditions was shown in **Figure 6B-C**. In the platelet rich plasma samples, 85.26±0.32% of platelets in the platelet rich plasma of DNAgel group were activated, which was second only to the TRAP-6 positive control (90.23±0.29%), and higher than the group of GS (78.73±3.81%), DNAaq (66.73±1.36%) and untreated (4.93±3.01%) groups. Moreover, 71.23±0.81% of the platelets in the whole blood of DNAgel group were activated, comparable to the TRAP-6 group (78.73±0.44%) and apparently higher than the other three groups.

Furthermore, according to the suggestions from the reviewer and the selection of commercial flow cytometry antibodies, we utilized human whole blood for analyzing the activation of GPIIb/IIIa, by using the specific antibodies of PE Anti-Human CD61 and FITC Anti-Human PAC-1, to measure the conformational change of GPIIb/IIIa from the low-affinity into the high-affinity state. The results were shown in **Figure S12** and the gating strategy was also provided. Actually, 11.20±0.17% of the platelets were PAC-1 positive in the DNAgel group, which was lower than that of the TRAP-1 group (34.17±1.74%), but significantly higher than that of the GS group (5.05±0.25%), DNAaq group (1.43±0.09%) and untreated group (0.97±0.14%). These results showed that the DNA hydrogel we prepared could induce a conformational change of GPIIb/IIIa on the surface of platelets from the low-affinity state into the high-affinity state, thereby promoting platelet activation and aggregation.

Figure S11. Representative flow cytometry gating strategy for FITC Mouse Anti-Rat CD61 and PE Mouse Anti-Rat CD62P stained cells in whole blood sample with TRAP-6 treated.

Figure 6. DNAGel promoted platelet activation and accelerated the clotting process by upregulating related genes. (A) Signaling pathway of clotting cascade triggered by DNAGel. (B) Statistical results of platelet activation and (C) Representative flow cytometry plots under different treatment conditions. Error bar, SDs ($n \geq 3$). (D) RNA-seq analysis of whole blood cells, and differential expressed genes presented in a volcano map. Cutoff line: Log_2 fold change (Log_2FC) $> |2|$ and adjusted p value < 0.01 . (E) Heatmap of differential expressed genes related to platelet activation and (F) GSEA of KEGG pathway analysis

related to platelet.

Figure S12. (A) Representative flow cytometry plots of whole blood samples under different treatment conditions. (B) Statistical results of GPIIb/IIIa in high-affinity or low-affinity state. Error bar, SDs ($n \geq 3$). (C) Exemplary gating strategy for PE Anti-Human CD61 and FITC Anti-Human PAC-1 stained cells in TRAP-6 treated group.

2) Instead of using proposed methods such as immunohistochemistry and suitable antibodies, the authors respond to this comment by referring again to the measurement of total GPIIb/IIIa by ELISA, which is not appropriate to answer the question raised. In addition, the authors did not comment on whether other cell types or blood clot components were bound to the DNA hydrogel.

Response: Thanks again for the professional and constructive suggestions. Here, we utilize method of immunofluorescence to visualize platelet activation. The antibody against P-selectin was used to stain activated platelets. It was observed in **Figure S5A** that activated platelets were clustered in the DNA network (blue), appearing red fluorescent. However, as for other cells like lymphocytes (CD3), monocytes (CD11b), and leucocytes (CD45), there were no specific staining patterns for these cells in **Figure S5B** (Labastide, W.B., et al. Blood, 1990; Dziennis, S., et al. Blood, 1995; Barford, D., et al. Science, 1994). Besides, a large amount of fibrin (white arrow) was observed in the blood clot (**Figure S4A**), while fibrin was also found to adhere to the surface of the DNAgel treated with platelet rich plasma sample (**Figure S4B**). In contrast, there was less blood cells, clots and fibrin adhered to the surface of GS when compared to that of the DNAgel group.

Figure S5. (A) CLSM images of activated platelets (red) aggregated and adhered to the surface of DNA network (blue). (B) lymphocytes (CD3), monocytes (CD11b), and leucocytes (CD45) were barely observed on the surface of the DNAgel (blue).

Figure S4. Representative SEM images of fibrin (white arrow) adhered to the surface of DNAgel and GS. (A) DNAgel treated with whole blood, (B) DNAgel treated with platelet rich plasma, (C) GS treated with whole blood, and (D) GS treated with platelet rich plasma.

4) Referring a third time to total GPIIb/IIIa content and the same data as before is not satisfying. However, the authors now present new data indirectly showing that FVII might bind to the DNA hydrogel but neglecting the activation status of FVII.

Response: We appreciate the valuable suggestions. Previously, we've deduced that FVII might adhere to the DNA hydrogel. Here, we quantified the activated Factor VII (FVIIa) in DNAgel-treated plasma by using Human Activated Coagulation Factor VIIa (FVIIa) ELISA Kit. As shown in **Figure S14**, more activated Factor VII was detected in the DNAgel-treated plasma, when compared to the untreated group. It was suggested that the negatively charged DNAgel may activate FSAP, which further activates the Factor VII.

FigureS14. FVIIa content in the DNAgel-treated plasma. Error bar, SDs (n=3)

7) Done for the figures of the initial version of the manuscript but not provided in newly added Figures S1 and S10-S14.

Response: Thank you for the kind reminder. We've indicated the number of experiments in the figure legends, and verified it throughout the text.

Reviewers' Comments:

Reviewer #1:

Remarks to the Author:

The authors tried to address the reviewer's concerns regarding the novelty of the materials and their adhesive strength compared clinically useful glue. Now, the manuscript is acceptable for publication.

Reviewer #2:

Remarks to the Author:

I have no further comments.